# Robust Generalization of Quadratic Neural Networks via Function Identification

## Abstract

A key challenge facing deep learning is that neural networks are often not robust to shifts in the underlying data distribution. We study this problem from the perspective of the statistical concept of parameter identification. Generalization bounds from learning theory often assume that the test distribution is close to the training distribution. In contrast, if we can identify the "true" parameters, then the model generalizes to arbitrary distribution shifts. However, neural networks are typically over-parameterized, making parameter identification impossible. We show that for quadratic neural networks, we can identify the function represented by the model even though we cannot identify its parameters. Thus, we can obtain robust generalization bounds even in the over-parameterized setting. We leverage this result to obtain new bounds for contextual bandits and transfer learning with quadratic neural networks. Overall, our results suggest that we can improve robustness of neural networks by designing models that can represent the true data generating process. In practice, the true data generating process is often very complex; thus, we study how our framework might connect to neural module networks, which are designed to break down complex tasks into compositions of simpler ones. We prove robust generalization bounds when individual neural modules are identifiable.

## 1 Introduction

Recent work has demonstrated that neural networks are not robust to shifts in the underlying data, including both distribution shifts (i.e., where the data comes from a new distribution independent of the neural network parameters) (Hendrycks & Dietterich, 2019; Taori et al., 2020) as well as adversarial shifts (i.e., where the shift can depend on the neural network parameters) (Szegedy et al., 2013). Accordingly, there has been a great deal of interest in better understanding why neural networks fail to be robust (Tsipras et al., 2018; Ilyas et al., 2019) and on improving robustness (Goodfellow et al., 2014; Raghunathan et al., 2018; Cohen et al., 2019).

From the perspective of learning theory, there is little reason to expect neural networks to be robust, since generalization bounds typically assume that the test examples are from the same distribution as the training examples. PAC-Bayesian generalization bounds allow for a limited amount of robustness, but only if the support of the target distribution $q$ is contained in that of the source distribution $p$, since it requires that the KL divergence $D_{\mathrm{KL}}(q \parallel p)$ is finite. Yet, distribution shifts (Hendrycks & Dietterich, 2019) often shift probability mass to inputs completely outside the source distribution.

Instead, the reason we might expect neural networks to be robust to these shifts is that humans are robust to them; for instance, small pixel-level shifts considered in adversarial examples are typically unnoticeable to humans, yet these shifts can move the image completely off of the distribution of natural images. This fact indicates a gap in our theoretical understanding of neural networks. In particular, the key question is understanding settings under which we may expect neural networks to be robust to distribution shifts that are "large" (e.g., in terms of KL divergence).

We study a strategy for closing this gap based on the statistical concept of *identifiability* (Hsu et al., 2012). At a high level, this concept assumes that the true model belongs to the model family; then, in the limit of infinite training data, the learning algorithm can exactly recover the parameters of the true model. For instance, in linear regression, the data is generated according to the model $y = \langle \theta^*, x \rangle + \xi$, where $\xi$ is $\sigma$-subgaussian noise. Then, under mild assumptions on the training data $Z = (X, Y)$, the

ordinary least squares (OLS) estimator $\hat{\theta}(Z)$ recovers the true parameter—i.e., in the limit of infinite data, $\hat{\theta}(Z) = \theta^*$. With finite samples, OLS satisfies high-probability convergence rates of the form

$$\|\hat{\theta}(Z) - \theta^*\|_2 \leq \epsilon. \tag{1}$$

The connection to robustness is that if (1) holds, then for *any* input $x$ such that $\|x\|_2 \leq x_{\max}$, we have

$$|\langle \hat{\theta}(Z), x \rangle - \langle \theta^*, x \rangle| \leq \|\hat{\theta}(Z) - \theta^*\|_2 \cdot \|x\|_2 \leq \epsilon \cdot x_{\max}. \tag{2}$$

Thus, for *any* distribution $q(x)$ with support on $B_2(0, x_{\max}) = \{x \in \mathcal{X} \mid \|x\|_2 \leq x_{\max}\}$, $\hat{\theta}(Z)$ obtains bounded error—i.e., we have $\mathbb{E}_{q(x)}[(\langle \hat{\theta}(Z), x \rangle - \langle \theta^*, x \rangle)^2] \leq \epsilon^2 x_{\max}^2$ with high probability.

Thus, a natural question is whether we can obtain similar kinds of parameter identification bounds for neural networks. A key complication is that practical neural networks are often *over-parameterized*, possibly to facilitate optimization (Du & Lee, 2018; Jacot et al., 2018). In this setting, identification is impossible, since multiple parameters can yield the same model. Nevertheless, it may be possible to obtain bounds of the form (2)—in particular, even if we do not recover the true parameters $\theta^*$, we can still recover the function $f_{\theta^*}(x)$. We refer to this notion as *function identification*. Furthermore, we show that quadratic neural networks satisfy function identification bounds under mild conditions.

To demonstrate its utility, we show how function identification can be leveraged to obtain regret guarantees for a bandit (Rusmevichientong & Tsitsiklis, 2010) where each arm is a quadratic neural network. Linear bandits fundamentally involve covariate shift since their "covariates" are arms, which are adaptively chosen through the learning process as a function of past observations; thus, existing approaches have all operated in the setting where there is a *unique* and identifiable global minimizer. Similarly, we build on recent work proving bounds on transfer learning in the setting of bounded label shift and unbounded covariate shift (Bastani, 2020; Xu et al., 2021); again, we show that we can leverage function identification to easily transfer learn quadratic neural networks.

Our results suggest that one strategy for improving robustness of neural networks is to design models that can represent the true data generating process. However, doing so can be challenging due to the complexity of most real-world data generating processes mapping covariates to labels—e.g., mapping natural language to semantic meaning or images to object detections and labels. As a consequence, we study how our results can connect to neural module networks (Andreas et al., 2016), which are designed to break down complex tasks into smaller ones that can each be solved by a neural network. For instance, we may break down the task "count the number of red balls in image $x$" into (i) detecting balls, (ii) detecting red objects, and (iii) intersecting the two sets, and (iv) summing the results.

Intuitively, neural modules can generalize more robustly since (i) it is more likely that an individual neural module designed to solve a simple task can be identified from training data, and (ii) even if module composition is not itself identifiable, shifts in the compositional structure tend to be smaller than shifts in the underlying data distribution. We study a simplified form of neural module networks, where modules are quadratic neural networks composed in sequence according to a given input; for simplicity, we assume they can be trained in a supervised way, ensuring robust generalization. Then, we show that under certain conditions, compositions of these models are also robust, including the case where there are shifts in the distribution over compositions.

**Related work.** Prior work has connected misspecification (i.e., the true model is in the model family) and robustness to covariate shift (Shimodaira, 2000; Wen et al., 2014); however, having a correctly specified model is insufficient if the true parameters are not identifiable—e.g., in linear regression, if the covariance matrix $\Sigma = \mathbb{E}_{p(x)}[xx^\top]$ is singular, then $\theta$ is not identifiable; thus, the estimated model may not be robust. Quadratic neural networks cannot be identified even if the model is correctly specified since the parameters have a continuous symmetry (i.e., orthogonal transformations).

Recent work has studied learning under adversarial examples (Goodfellow et al., 2014; Raghunathan et al., 2018; Cohen et al., 2019) and corrupted training data (Steinhardt et al., 2017; Diakonikolas et al., 2019). In contrast, we are interested in robustness to covariate shift; there has been recent work empirically showing that neural networks are sensitive to distribution shift (Hendrycks & Dietterich, 2019; Taori et al., 2020; Ruis et al., 2020; Ribeiro et al., 2020; Koh et al., 2020). Distributionally robust optimization enables training of models robust to small shifts (Duchi & Namkoong, 2018), but we are interested in potentially large shifts. Unsupervised domain adaptation (Ben-David et al., 2007; Blitzer et al., 2008) learns a model on a covariate shifted target distribution; however, they rely on unlabeled examples from the target domain, whereas we do not. There has been recent theory on

robustness to adversarial perturbations—e.g., showing there may be a tradeoff between robustness and on-distribution generalization (Tsipras et al., 2018), and that non-robust algorithms tend to learn predictive but brittle representations compared to adversarially robust ones (Ilyas et al., 2019). In contrast, we show that these tradeoffs are mitigated when the true model function can be identified despite over-parameterization. Furthermore, adversarial shifts are typically bounded (e.g., small $\ell_\infty$ norm), whereas the shifts we consider may be large.

There has been a great deal of recent work on deep learning theory, including on quadratic neural networks; however, it has largely focused on optimization (Ge et al., 2017b; Jacot et al., 2018; Du et al., 2019; Gao et al., 2019; Soltanolkotabi et al., 2018; Li et al., 2018), and on-distribution generalization (Neyshabur et al., 2017; Du & Lee, 2018; Jacot et al., 2018; Arora et al., 2018; Long & Sedghi, 2019). In contrast, we are interested in out-of-distribution generalization.

We discuss additional related work on matrix factorization and multi-armed bandits in Appendix A, as well as a discussion of the novelty of our results.

## 2 PROBLEM FORMULATION

We consider a model $f_\theta : \mathcal{X} \to \mathcal{Y}$, with covariates $\mathcal{X} \subseteq \mathbb{R}^d$, labels $\mathcal{Y} \subseteq \mathbb{R}$, and parameters $\theta \in \Theta \subseteq \mathbb{R}^m$. A generalization bound from learning theory typically has the form

$$\mathbb{P}_{p(Z)}[L_p(\hat{\theta}(Z)) \le \epsilon] \ge 1 - \delta \qquad \text{where} \qquad L_p(\theta) = \mathbb{E}_{p(x)}[(f_\theta(x) - f_{\theta^*}(x))^2], \qquad (3)$$

where $\epsilon, \delta \in \mathbb{R}_{>0}$, $Z = \{(x_1, y_1), ..., (x_n, y_n)\} \subseteq \mathcal{X} \times \mathcal{Y}$ with $y_i = f_{\theta^*}(x_i) + \xi_i$ is a training set of i.i.d. observations from a distribution $p$ (i.e., $p(Z) = p(x_1, y_1) \cdot ... \cdot p(x_n, y_n)$), $\xi_i$ is bounded random noise independent of $x_i$ with $|\xi_i| \le \xi_{\max}$, $\theta^* \in \Theta$ are the true parameters, and

$$\hat{\theta}(Z) = \arg\min_{\theta \in \Theta} \hat{L}(\theta; Z) \qquad \text{where} \qquad \hat{L}(\theta; Z) = \frac{1}{n} \sum_{i=1}^n (f_\theta(x_i) - y_i)^2$$

is an estimator based on the training data $Z$.[1] In particular, they assume that the training inputs $x_i \sim p$ are i.i.d. samples from the same distribution as the test example $x \sim p$.

**Definition 2.1.** The model $f_\theta$ and distribution $p$ satisfy *function identification* if for any $\epsilon, \delta \in \mathbb{R}_{>0}$, we have $\mathbb{P}_{p(Z)}[\forall x \in \mathcal{X} . (f_{\hat{\theta}(Z)}(x) - f_{\theta^*}(x))^2 \le \epsilon] \ge 1 - \delta$ for $n = |Z|$ sufficiently large.

Function identification implies generalization bounds even when the test data comes from a different distribution $q$. In particular, we say $f_\theta$ *robustly generalizes* if for any $q$ with support on $\mathcal{X}$, we have

$$\mathbb{P}_{p(Z)}[L_q(\hat{\theta}(Z)) \le \epsilon] \ge 1 - \delta, \qquad (4)$$

where the difference from (3) has been highlighted in red. It is easy to see that function identification implies (4). Note that the true model $f_{\theta^*}$ does not change, so there is no label shift.

## 3 FUNCTION IDENTIFICATION OF QUADRATIC NEURAL NETWORKS

Traditional statistical bounds on parameter identification can provide guarantees for arbitrary covariate shift. In particular, suppose we have a bound of the form

$$\mathbb{P}_{p(Z)}\left[\|\hat{\theta}(Z) - \theta^*\|_2 \le \epsilon\right] \ge 1 - \delta, \qquad (5)$$

and assume that the model family $f_\theta$ is $K$-Lipschitz continuous in $\theta$; then, we have

$$L_q(\hat{\theta}(Z)) \le K^2 \cdot \|\hat{\theta}(Z) - \theta^*\|_2^2 \le K^2 \epsilon^2 \qquad (6)$$

with probability at least $1 - \delta$ according to $p(Z)$. In particular, this bound holds for any covariate distribution $q$. Our goal is to extend these techniques to quadratic neural networks, which are over-parameterized so we cannot identify the true parameters $\theta^*$—i.e., (5) does not hold.

---

[1]Note that in (3), the loss $L_p$ omits the label errors $\xi$; including it would result in an additive constant to $L_p$. This choice ensures that the optimal parameters have zero loss—i.e., $L_q(\theta^*) = 0$ for any $q$.

### 3.1 QUADRATIC NEURAL NETWORKS

We consider a neural network $f_\theta$, where $\theta \in \mathbb{R}^{d \times k}$, with a single hidden layer with $k$ neurons—i.e., $f_\theta(x) = \sum_{j=1}^k a_j \cdot \sigma(\langle \theta_j, x \rangle)$. We consider the over-parameterization case where $k$ can be much larger than $d$. Following prior work (Du & Lee, 2018), we assume that $f_\theta$ has quadratic activations and output weights equal to one—i.e., $\sigma(z) = z^2$ and $a_j = 1$ for each $j \in [k]$, so we have

$$f_\theta(x) = \sum_{j=1}^k \langle \theta_j, x \rangle^2.$$

We assume the true (training) loss is the mean-squared error $L_p(\theta) = \mathbb{E}_{p(x)}[(f_\theta(x) - f_{\theta^*}(x))^2]$, and we consider a model trained using an empirical estimate of this loss on the training dataset:

$$\hat\theta(Z) = \arg\min_{\theta \in \Theta} \hat{L}(\theta; Z) \qquad \text{where} \qquad \hat{L}(\theta; Z) = \frac{1}{n} \sum_{i=1}^n (f_\theta(x_i) - y_i)^2.$$

Now, our goal is to obtain a bound of the form (6); to this end, we assume the following:

**Assumption 3.1.** $\|x\|_2 \le x_{\max}$ and $\|\theta\|_F \le \theta_{\max}$.

**Assumption 3.2.** There exists $\alpha \in \mathbb{R}_{>0}$ such that $\mathbb{E}_{p(x)}[(x^\top \Delta x)^2] \ge \alpha \|\Delta\|_F^2$ for any symmetric $\Delta \in \mathbb{R}^{d \times d}$.

Our second assumption is standard; in particular, it is closely related to the assumption in linear regression that the minimum eigenvalue of the covariance matrix is lower bounded—i.e., $\Sigma = \mathbb{E}_{p(x)}[xx^\top] \succ 0$. As an example, when $x$ is i.i.d. uniform in each component, e.g., $p(x) = \prod_{i=1}^d \text{Uniform}(x_i; [-1/2, 1/2])$, then we can take $\alpha = 1/180$; we give a proof in Appendix B.1.

### 3.2 ROBUST GENERALIZATION

Our approach leverages the fact that $f_\theta(x) = x^\top (\theta \theta^\top) x$; thus, $f_\theta$ resembles a matrix factorization model. Recent work has leveraged this connection to translate matrix factorization theory to quadratic neural networks (Du & Lee, 2018). We let $g(\theta) = \theta \theta^\top$ and $\tilde{f}_\phi(x) = x^\top \phi x$, where $\phi \in \Phi \subseteq \mathbb{R}^{d \times d}$, in which case $f_\theta(x) = \tilde{f}_{g(\theta)}(x)$; in addition, we define $\tilde{L}_p(\phi) = \mathbb{E}_{p(x)}[(\tilde{f}_\phi(x) - \tilde{f}_{\phi^*}(x))^2]$, where $\phi^* = g(\theta^*)$, and $\hat{\tilde{L}}(\phi; Z) = n^{-1} \sum_{i=1}^n (\tilde{f}_\phi(x_i) - y_i)^2$, so $L_p(\theta) = \tilde{L}_p(g(\theta))$ and $\hat{L}(\theta; Z) = \hat{\tilde{L}}(g(\theta); Z)$. Finally, we assume that $\|\phi\|_F \le \phi_{\max}$; in general, we have $\phi_{\max} \le \theta_{\max}^2$ by Assumption 3.1.

We begin by stating several lemmas establishing the properties needed for function identification. Our first lemma says that the loss is strongly convex in $\phi$.

**Lemma 3.3.** *Under Assumption 3.2, the loss $\tilde{L}_p(\phi)$ is $2\alpha$-strongly convex in $\phi$.*

We give a proof in Appendix B.2. Our next lemma says that our model family is Lipschitz in $\phi$.

**Lemma 3.4.** *Under Assumptions 3.1 & 3.2, $\tilde{f}_\phi$ and $\tilde{L}$ are $K$-Lipschitz in $\phi$, where $K = 4\phi_{max} x_{max}^4$.*

We give a proof in Appendix B.3. Our final lemma says that our estimate of the loss function is a uniformly good approximation of the true loss.

**Lemma 3.5.** *Under Assumptions 3.1 & 3.2, for any $\delta \in \mathbb{R}_{>0}$, we have*

$$\mathbb{P}_{p(Z)} \left[ \sup_{\theta \in \Theta} |L_p(\theta) - \hat{L}(\theta; Z) - \sigma(Z)| \le \epsilon \right] \ge 1 - \delta,$$

*where $\sigma(Z) = n^{-1} \sum_{i=1}^n \xi_i^2$, and letting $\ell_{max} = 2x_{max}^2 \phi_{max}$ be an upper bound on $|f_\theta(x) - f_{\theta^*}(x)|$,*

$$\epsilon = \sqrt{\frac{18\ell_{max}^2 (\ell_{max}^2 + \xi_{max}^2)}{n} \left( d^2 \max\left\{ 1, \log\left( 1 + \frac{8\phi_{max} K n}{\ell_{max}^2} \right) \right\} + \log \frac{2}{\delta} \right)}. \tag{7}$$

We give a proof in Appendix B.4. Note that $\epsilon \to 0$ as $n \to \infty$. Next, we prove our main result, which says that quadratic neural networks can be functionally identified.

**Theorem 3.6.** *Under Assumptions 3.1 & 3.2, we have*

$$\mathbb{P}_{p(Z)}\left[\forall x \in \mathcal{X} \, . \, (f_{\hat{\theta}(Z)}(x) - f_{\theta^*}(x))^2 \le \frac{2K^2\epsilon}{\alpha}\right] \ge 1 - \delta.$$

*Proof.* By Lemma 3.3, and since $\nabla_\phi \tilde{L}(g(\theta^*)) = 0$, we have

$$L_p(\hat{\theta}(Z)) - L_p(\theta^*) = \tilde{L}_p(g(\hat{\theta}(Z))) - \tilde{L}_p(g(\theta^*)) \ge \alpha\|g(\hat{\theta}(Z)) - g(\theta^*)\|_F^2. \tag{8}$$

Next, by Lemma 3.5 and the fact that $\hat{\theta}$ minimizes $\hat{L}(\theta; Z)$, we have

$$L_p(\hat{\theta}(Z)) \le \hat{L}(\hat{\theta}; Z) + \epsilon - \sigma(Z) \le \hat{L}(\theta^*; Z) + \epsilon - \sigma(Z) \le L_p(\theta^*) + 2\epsilon \tag{9}$$

with probability at least $1 - \delta$. Combining (8) and (9), we have

$$\|g(\hat{\theta}(Z)) - g(\theta^*)\|_F \le \sqrt{\frac{2\epsilon}{\alpha}}$$

with probability at least $1 - \delta$. Finally, by Lemma 3.4, we have

$$(f_{\hat{\theta}(Z)}(x) - f_{\theta^*}(x))^2 = (\tilde{f}_{g(\hat{\theta}(Z))}(x) - \tilde{f}_{g(\theta^*)}(x))^2 \le K^2\|g(\hat{\theta}(Z)) - g(\theta^*)\|_2^2 \le \frac{2K^2\epsilon}{\alpha} \; (\forall x \in \mathcal{X})$$

with probability at least $1 - \delta$, as claimed. $\square$

As a result, we provide a robust generalization error bound for quadratic neural networks with potential distribution shifts.

**Corollary 3.7.** *Under Assumptions 3.1 & 3.2, for any distribution $q(x)$ with support on $B_2(0, x_{max})$,*

$$\mathbb{P}_{p(Z)}\left[L_q(\hat{\theta}(Z)) \le \frac{2K^2\epsilon}{\alpha}\right] \ge 1 - \delta.$$

Finally, we also prove that gradient descent can find the global minima of $\hat{L}(\theta; Z)$, which ensures that gradient descent can perform function identification in practice; we give a proof in Appendix B.5.

**Proposition 3.8.** *All local minima of $\hat{L}(\theta; Z)$ are also global minima.*

## 4 QUADRATIC NEURAL BANDITS

A key application of robust generalization bounds is to parametric bandits; this is because, in bandit learning, the distribution of inputs $x$ used to estimate $\hat{\theta} \approx \theta^*$ can differ from the distribution under which $f_{\hat{\theta}}$ is used. Thus, generalization bounds based on notions such as Rademacher complexity cannot be used. Unlike prior literature in bandits, we consider an over-parameterized function that does *not* admit a unique solution; in contrast, recent work on neural tangent kernel bandits (Zhou et al., 2020) assumes that there is a unique, identifiable solution. Note that this assumption cannot hold for quadratic neural networks because they are invariant to transformations such as rotations.

We consider a standard linear bandit (Rusmevichientong & Tsitsiklis, 2010; Abbasi-Yadkori et al., 2011) with a fixed horizon $T \in \mathbb{N}$, but where the expected reward is parameterized by a quadratic neural network instead of a linear function. At each time step $t$, the algorithm chooses among a continuum of actions $x_t \in \mathcal{X}$, and receives a reward

$$y_t = f_{\theta^*}(x_t) + \xi_t = \sum_{j=1}^{k} \langle \theta_j^*, x_t \rangle^2 + \xi_t, \tag{10}$$

where $\theta^* \in \mathbb{R}^{d \times k}$ is an unknown parameter matrix, and $\xi_t$ are bounded i.i.d. random variables. For simplicity, we assume that $\mathcal{X} = B_2(0, 1)$ is the unit ball. Then, our goal is to bound the *regret*

$$R(T) = \sum_{t=1}^{T} (\mathbb{E}_{p(\xi_t)}[y_t] - y^*) \qquad \text{where} \qquad y^* = \max_{x \in \mathcal{X}} f_{\theta^*}(x).$$

We make the following assumption, which says that $\phi^* = \theta^*\theta^{*\top}$ has a gap in its top eigenvalue:

---

**Algorithm 1** Explore-Then-Commit Algorithm for Quadratic Neural Network Bandit

---

**procedure** QUADRATICNEURALBANDIT
    Initialize $Z \leftarrow \varnothing$
    Let $m$ be as in (11)
    **for** $t \in \{1, ..., m\}$ **do**
        Sample i.i.d. action $x_t \sim p$, where $p$ is as in (12)
        Take action $x_t$ and obtain reward $y_t$ as in (10)
        Update $Z \leftarrow Z \cup \{(x_t, y_t)\}$
    **end for**
    Compute $\hat{\theta} = \arg\min_\theta \hat{L}(\theta; Z)$, where $\hat{L}(\theta; Z) = m^{-1} \sum_{i=1}^m (f_\theta(x_i) - y_i)^2$
    Compute $\hat{x} = \arg\max_{x \in \mathcal{X}} f_{\hat{\theta}}(x)$
    **for** $t \in \{m+1, ..., T\}$ **do**
        Take action $x_t = \hat{x}$ and obtain reward $y_t$ as in (10)
    **end for**
**end procedure**

---

**Assumption 4.1.** Let $\phi^* = \theta^* \theta^{*\top}$, and let $\lambda_1 \geq \lambda_2 \geq ... \geq \lambda_d$ be the eigenvalues of $\phi^*$. There exists a constant $M \in \mathbb{R}_{>0}$ such that $\lambda_1 - \lambda_2 \geq 4/M$.

This assumption ensures that the eigenvectors of $\phi^*$ to be stable under perturbations. The eigenvectors of $\phi^*$ correspond to the optimal action $x^* = \arg\max_{x \in \mathcal{X}} f_{\theta^*}(x)$ since $f_{\theta^*}(x) = x^\top \phi^* x$; thus, this assumption ensures that if $\hat{\theta} \approx \theta^*$, then the optimal action $\hat{x} = \arg\max_{x \in \mathcal{X}} f_{\hat{\theta}}(x)$ satisfies $\hat{x} \approx x^*$.

Next, we describe our algorithm, summarized in Algorithm 1. We consider an explore-then-commit strategy for simplicity, since it already achieves the asymptotically optimal regret rate (Rusmevichientong & Tsitsiklis, 2010); our approach can similarly be applied to more sophisticated algorithms such as UCB (Abbasi-Yadkori et al., 2011) and Thompson sampling (Agrawal & Goyal, 2013). Our algorithm proceeds in two stages: (i) the *exploration stage* (for $t \in \{1, ..., m\}$), and (ii) the *exploitation stage* (for $t \in \{m+1, ..., T\}$), where

$$m = \left\lceil \left( \frac{135 M (\ell_{\max} + \xi_{\max})^2 d^3 T \sqrt{\log(3 + 8\phi_{\max} KT/\ell_{\max}^2)}}{\phi_{\max}} \right)^{2/3} \right\rceil. \tag{11}$$

In the exploration stage, we randomly choose actions $x_t \sim p$, where

$$p(x) = \prod_{i=1}^d \text{Uniform}\left( x_i; \left[ -\frac{1}{\sqrt{d}}, \frac{1}{\sqrt{d}} \right] \right). \tag{12}$$

Note that $\|x\|_2 \leq 1$ for $x$ in the support of $p$, so $x_t \in \mathcal{X}$. Following the discussion in Section 3, for this choice of $p$, Assumption 3.2 holds for the dataset $Z$ with $\alpha = 4/(45d^2)$.

Next, we compute an estimate $\hat{\theta}$ of $\theta^*$ based on the data $Z$ collected so far, and compute the optimal action $\hat{x}$ assuming $\hat{\theta}$ are the true parameters. Then, in the exploitation stage, we always use action $\hat{x}$.

The key challenge providing theoretical guarantees using traditional generalization bounds is handling the optimization problem over $x \in \mathcal{X}$ used to compute $\hat{x}$. Since $\hat{x}$ is not sampled from the distribution $p$, traditional bounds do not provide any guarantees about the accuracy of $f_{\hat{\theta}}(\hat{x})$ compared to $f_{\theta^*}(\hat{x})$. In contrast, Theorem 3.6 provides a uniform guarantee, and can therefore be used to bound the regret.

**Theorem 4.2.** *Under Assumptions 3.1, 3.2 & 4.1, the expected regret of Algorithm 1 is*

$$R(T) \leq C_0 + C_1 \cdot T^{2/3} \left( \log\left( 3 + \frac{8\phi_{max} KT}{\ell_{max}^2} \right) \right)^{1/3},$$

*where $C_0$ and $C_1$ do not depend on $T$ (see Appendix C).*

In particular, we have $R(T) = \tilde{O}(T^{2/3})$; we give a proof in Appendix C. Note that this is worse than the usual $\tilde{O}(\sqrt{T})$ regret because Theorem 3.6 only admits a $n^{1/4}$ convergence rate.

## 5 TRANSFER LEARNING OF QUADRATIC NEURAL NETWORKS

So far, we have considered shifts in the covariate distribution but not in the label distribution. Now, we consider a transfer learning problem where there is additionally a small shift in the labels. In particular, we assume we have *proxy data* $Z_p \subseteq \mathcal{X} \times \mathcal{Y}$ from the source domain of the form $y_{p,i} = f_{\theta_p^*}(x_{p,i}) + \xi_{p,i}$ (for $i \in [n_p]$), where $\theta_p^* \in \Theta$ are the proxy parameters and $p(x_p)$ is the source covariate distribution, along with *gold data* $Z_g \subseteq \mathcal{X} \times \mathcal{Y}$ from the target domain of the form $y_{g,i} = f_{\theta_g^*}(x_{g,i}) + \xi_{g,i}$ (for $i \in [n_g]$), where $\theta_g^* \in \Theta$ are the gold parameters and $q(x_g)$ is the target covariate distribution. We are interested in the setting $n_p \gg n_g$, and where $\|\theta_g^* - \theta_p^*\|_F \leq B$ is small.

We consider a two-stage estimator (Bastani, 2020) that first computes an estimate of the proxy parameters $\hat{\theta}_p = \arg\min_{\theta \in \Theta} \hat{L}(\theta; Z_p)$, and then computes an estimate of the gold parameters in a way that is constrained towards the proxy parameters. First, note that we have

$$\mathbb{P}_{p(Z)}\left[L_q(\hat{\theta}_p) \leq \frac{2K^2\epsilon_p}{\alpha}\right] \geq 1 - \frac{\delta}{2},$$

where

$$\epsilon_p = \sqrt{\frac{18\ell_{\max}^2(\ell_{\max}^2 + \xi_{\max}^2)}{n_p}\left(d^2 \max\left\{1, \log\left(1 + \frac{8\phi_{\max}Kn_p}{\ell_{\max}^2}\right)\right\} + \log\frac{4}{\delta}\right)},$$

where we have highlighted the differences from $\epsilon$ in (7) in red. Next, we make a technical assumption:

**Assumption 5.1.** For some $\sigma_0 \in \mathbb{R}_{>0}$, $\sigma_{\min}(\theta_p^*) \geq \sigma_0$, where $\sigma_{\min}(\theta)$ is the $d$th singular value of $\theta$.

Equivalently, the minimum eigenvalue of $g(\theta_p^*)$ is positive; intuitively, this assumption ensures a good estimate of $\theta_p^*\theta_p^{*\top}$ implies a good estimate of $\theta_p^*$ (up to an orthogonal transformation). Then, letting

$$\hat{B} = B + \frac{1}{\sigma_0}\sqrt{\frac{2\epsilon_p}{\alpha}}$$

be an expanded radius to account for error in our estimate of $\hat{\theta}_p$, we use the estimator

$$\hat{\theta}_g = \arg\min_{\theta \in B_2(\hat{\theta}_p, \hat{B})} \hat{L}(\theta; Z_g) \qquad \text{where} \qquad B_2(\hat{\theta}_p, \hat{B}) = \{\theta \in \Theta \mid \|\theta - \hat{\theta}_p\|_F \leq \hat{B}\}.$$

Note that we have assume $\hat{B}$ is known; in practice, this constraint can be included as an additive regularization term. Intuitively, this formulation mirrors transfer learning algorithms based on fine-tuning—i.e., initializing the parameters to the proxy data $\theta_p$ and then taking a small number of steps of stochastic gradient descent (SGD) on the gold data $\theta_g$. In particular, SGD can be interpreted as $L_2$ regularization on the parameters (Ali et al., 2020), so fine-tuning $L_2$-regularizes $\hat{\theta}_g$ towards $\theta_p$.

**Theorem 5.2.** *Under Assumptions 3.1, 3.2 & 5.1, for any $q(x)$ with support on $B_2(0, x_{max})$,*

$$\mathbb{P}_{p(Z)}\left[L_q(\hat{\theta}_g) \leq \frac{2K^2\epsilon_g}{\alpha}\right] \geq 1 - \delta,$$

*where*

$$\epsilon_g = \hat{B} \cdot \sqrt{\frac{18K^2(K^2\hat{B}^2 + \xi_{max}^2)}{n_g}\left(d^2 \max\left\{1, \log\left(1 + \frac{8\phi_{max}Kn_g}{\ell_{max}^2}\right)\right\} + \log\frac{4}{\delta}\right)}.$$

Thus, if $B$ is small and $n_p$ is large, $f_{\hat{\theta}_g}$ is accurate even if $n_g$ is small; we give a proof in Appendix D.

## 6 GENERALIZATION BOUNDS FOR NEURAL MODULE NETWORKS

While function identification enables robust generalization, many data generating processes are too complex to be identifiable. Neural module networks are designed to break complex prediction problems into smaller tasks that are individually easier to solve. These models take two kinds of input: (i) a sequence of tokens $w$ (e.g., word embeddings) indicating the correct composition of modules,

and (ii) the input $x$ to the modules. Then, the model predicts the sequence of modules $j_1...j_T$ based on $w$, and runs the modules in sequence to obtain output $x' = f_{j_T}(...(f_{j_1}(x))...)$.

We study conditions under which neural module networks can robustly generalize. Rather than study arbitrary distribution shifts, we consider two separate shifts:

- **Module inputs:** We assume that the individual modules are identifiable; as a consequence, we assume the shift to the module input $x$ can be arbitrary.

- **Module composition:** We consider shifts to the token sequence $w$. If the model mapping $w$ to $j_1...j_T$ is identifiable, then the entire model is identifiable. Instead, we show that when this model is not identifiable, compositional structure can still aid generalization. Intuitively, we show that while small shifts in the compositional structure can cause large shifts in the distribution $p(w)$, models that leverage the structure of $p(w)$ can still generalize well.

In more detail, consider a simplified neural module network $f$, which includes (i) a set of neural modules $\{f_j : \mathcal{X} \to \mathcal{X}\}_{j=1}^k$, and (ii) a parser $g : \mathcal{Z}^T \to [k]^T$, where $\mathcal{Z} \subseteq \mathbb{R}^r$, with model class $g \in \mathcal{G}$. We assume each component of $f_j(x)$ is computed by a separate quadratic neural network; we discuss the architecture of $g$ below. Then, given an input $x \in \mathcal{X} \subseteq \mathbb{R}^d$ and $w \in \mathcal{W} = \mathcal{Z}^T$, the corresponding neural module network $f : \mathcal{X} \times \mathcal{W} \to \mathcal{X}$ is defined by

$$f(x, w) = (f_{j_T} \circ ... \circ f_{j_1})(x) = f_{j_T}(...(f_{j_1}(x))...) \qquad \text{where} \qquad j_1...j_T = g(w).$$

We assume that $g$ has compositional structure—i.e., for some $\tilde{g} : [k] \times \mathcal{Z} \to [k]$, we have

$$g(w) = j_1...j_T \qquad \text{where} \qquad j_t = \begin{cases} 0 & \text{if } t = 0 \\ \tilde{g}(z_t, j_{t-1}) & \text{otherwise,} \end{cases}$$

where $w = z_1...z_T$. Intuitively, $w$ is a sequence of word vectors; then, the current neural module $j_t = \tilde{g}(z_t, j_{t-1})$ depends both on the current word vector $z_t$ and the previous neural module $j_{t-1}$. First, we assume that the individual modules have been functionally identified.

We assume we have fully labeled data we can use to train the neural modules—i.e., for each input $x$ and sequence $w$, we have both the desired sequence $j_1...j_T$ of neural modules, as well as the entire execution $x_0, x_1, ..., x_T$, where $x_0 = x$ and $x_{t+1} = f_{j_t}(x_t)$ otherwise. Thus, we can use supervised learning to train the neural modules;[2] in particular, we can construct labeled examples $(j_{t-1}, z_t, j_t)$ used to train the parser $\tilde{g}$, and labeled examples $(x_t, x_{t+1})$ to train the modules $f_{j_t}$. For simplicity, we assume we have a uniform lower bound $n$ on the number of training examples for the parser and for each module. Then, we have the following straightforward result:

**Lemma 6.1.** *Under Assumptions 3.1 & 3.2, with probability at least $1 - dk\delta$, for each $j \in [k]$,*

$$\|\hat{f}_j(x) - f_j^*(x)\|_2 \leq \sqrt{\frac{2dK^2\epsilon}{\alpha}} =: \epsilon_f \qquad (\forall x \in \mathcal{X}),$$

*where $\hat{f}_j$ is the estimated module and $f_j^*$ is the ground truth module.*

This result follows straightforwardly from Theorem 3.6 along with a union bound. In contrast, we do not assume the parsing model robustly generalizes, but only on distribution. For the subsequent analysis, we can use any neural network models that satisfy the statement of Lemma 6.1.

**Lemma 6.2.** *Under Assumptions 3.1 & 3.2, with probability at least $1 - \delta$, we have*

$$\mathbb{P}_{\tilde{p}(z,j)}\left[\hat{\tilde{g}}(z, j) \neq \tilde{g}^*(z, j)\right] \leq 4R_n(\mathcal{G}) + \sqrt{\frac{2\log(2/\delta)}{n}} =: \epsilon_g,$$

*where $R_n(\mathcal{G})$ is the Rademacher complexity of $\mathcal{G}$ (including its loss function), where $p(z, j) = T^{-1}\sum_{t=1}^T p_t(z, j)$, and where*

$$\tilde{p}_t(z, j) = \begin{cases} \mathbb{1}(j = 0) \cdot \tilde{p}(z) & \text{if } t = 1 \\ \sum_{j'=1}^k \int \mathbb{1}(j = \tilde{g}^*(z', j')) \cdot \tilde{p}(z \mid z') \cdot \tilde{p}_{t-1}(z', j')dz' & \text{otherwise.} \end{cases}$$

---

[2]Neural modules are often trained with only partial supervision (Andreas et al., 2016); we leave an analysis of this strategy to future work since our focus is on understanding generalization rather than learning dynamics.

This result is a standard Rademacher generalization bound (Bartlett & Mendelson, 2002). Note that we have also assumed that the distribution over token sequences is structured, which is necessary for our compositional implementation of $g$ to generalize, even on distribution. Intuitively, the distribution over $(z, j)$ consists of both a unigram model over the word vectors:

$$p(z_1, ..., z_T) = \prod_{t=1}^{T} \tilde{p}(z_t \mid z_{t-1}),$$

where we define $\tilde{p}(z_1 \mid z_0) = \tilde{p}(z_1)$, as well as a unigram model over neural modules:

$$p(j_1...j_T \mid z_1, ..., z_T) = \prod_{t=1}^{T} \mathbb{1}(j_t = \tilde{g}^*(z_t, j_{t-1})).$$

Next, we consider a shifted distribution $\tilde{q}(z \mid z')$, which is close to $\tilde{p}(z \mid z')$.

**Assumption 6.3.** We have $\|\tilde{q}(\cdot \mid z') - \tilde{p}(\cdot \mid z')\|_{\text{TV}} \leq \alpha$.

Importantly, despite this assumption, the shift between the overall distributions $p(z_1, ..., z_T)$ and $q(z_1, ..., z_T)$ can still be large since it compounds exponentially across the steps $t \in [T]$.

**Proposition 6.4.** *There exist $p$ and $q$ that satisfy Assumption 6.3, but $\|p - q\|_{TV} = 2(1 - (1 - \alpha/2)^T)$.*

That is, even if the single step probabilities $\tilde{p}(z \mid z')$ and $\tilde{q}(z \mid z')$ have total variation (TV) distance bounded as in Assumption 6.3, the overall distributions $p$ and $q$ can have TV distance exponentially close to the maximum possible distance of 2 in $T$; we give a proof in Appendix E.1.

We show that neural module networks generalize since $\hat{g}$ leverages the compositional structure of $p$. First, we show that under Assumption 6.3, the overall shift in the input distribution of $\hat{\tilde{g}}$ is bounded:

**Lemma 6.5.** *Under Assumptions 3.1, 3.2 & 6.3, we have $\|\tilde{q} - \tilde{p}\|_{TV} \leq T\alpha$, where $\tilde{p}$ is defined in Lemma 6.2 and $\tilde{q}$ is defined in Assumption 6.3.*

That is, while the shift can compound across steps $t$, it does so only linearly; we give a proof in Appendix E.2. Next, we show that as a consequence, the error of $\hat{g}$ is bounded.

**Lemma 6.6.** *Under Assumptions 3.1, 3.2 & 6.3, and assuming that $\mathbb{P}_{p(z,j)}[\hat{\tilde{g}}(z, j) \neq \tilde{g}^*(z, j)] \leq \epsilon_g$, we have that $\mathbb{P}_{p(w)}[\hat{g}(w) \neq g^*(w)] \leq T\epsilon_g$.*

We give a proof in Appendix E.3. Finally, we have our main result.

**Theorem 6.7.** *Under Assumptions 3.1, 3.2 & 6.3, with probability at least $1 - (dk + 1)\delta$, we have*

$$\mathbb{P}_{q(w)} \left[ \|\hat{f}(x, w) - f^*(x, w)\|_2 \leq T\epsilon_f \cdot \max\{K^{T-1}, 1\} \right] \geq 1 - T\epsilon_g - T^2\alpha.$$

We give a proof in Appendix E.4. Intuitively, Theorem 6.7 says that the error of the neural module network is linear in $T$ as long as $K \leq 1$. Note that even if there is no distribution shift, its error is

$$\mathbb{P}_{p(w)} \left[ \|\hat{f}(x, w) - f^*(x, w)\|_2 \leq T\epsilon_f \cdot \max\{K^{T-1}, 1\} \right] \geq 1 - T\epsilon_g,$$

by the same argument as the proof of Theorem 6.7. The exponential dependence on $K$ is unavoidable since $K > 1$ says that the modules $f_j$ can expand the input, which leads to exponential blowup in the magnitude of the output as a function of $T$, which also makes the estimation error exponential in $T$. Thus, the only cost to the distribution shift from $p$ to $q$ is the additional error probability $T^2\alpha$.

## 7 CONCLUSION

We have presented a number of results demonstrating that over-parameterization does not fundamentally harm learning models that are robust to arbitrary distribution shifts. In particular, even though we can no longer identify the true parameters for quadratic neural networks, we show that we can identify the true function, thereby enabling us to prove new results in bandits and transfer learning. Finally, we provide preliminary analysis extending these results to neural module networks to handle complex data generating processes. A limitation of our work is that it is specialized to quadratic neural networks; a key direction for future work is to explore how our results extend to other activation functions and deeper architectures.

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

# A  ADDITIONAL RELATED WORK

**Low-rank matrix factorization.** Our notion of functional identification for quadratic neural network is related to the low-rank matrix factorization literature. However, they impose a low-rank structure on the matrix to recover, and hence typically require extra conditions to identify the matrix—e.g., the restricted isometry property (RIP) (Candes & Plan, 2011; Ge et al., 2017a), or bounded $\ell_2$ norm of noise vector (Kabanava et al., 2016). In contrast, we consider a more general case and do not assume any underlying structure of the matrix; in particular, since our goal is to capture over-parameterization of neural networks, our matrix is usually decomposed as $\phi = \theta\theta^\top$, where $\theta \in \mathbb{R}^{d \times k}$ and $k \geq d$ (and $\phi$ is not necessarily low-rank). Also, we study the prediction error of neural networks in the presence of distribution shifts, whereas the goal of the low-rank literature is to recover the true matrix.

**Multi-armed bandits.** Prior literature on parameterized bandits has considered a number of functional forms, ranging from linear (Abbasi-Yadkori et al., 2011; Rusmevichientong & Tsitsiklis, 2010) to neural tangent kernels (Zhou et al., 2020). Most of this work makes a *realizability* assumption that the model family contains the true model;[3] implicitly, they consider model families where there is a *unique*, identifiable true parameter. These assumptions are necessary precisely due to the fact that the test and training distributions are different; thus, much of the bandit literature has focused on proving parameter identification results to enable learning. In contrast, the identifiability assumption does not hold for quadratic neural networks because they are invariant to parameter transformations. To the best of our knowledge, we consider the first over-parameterized bandit problem that considers a model that is not parameter-identifiable; we find that similar regret results hold as long as the function represented by the model can be identified. Separately, Foster & Rakhlin (2020) makes a general connection between online regression oracles and the regret of a bandit algorithm; however, their approach only provides good guarantees when the regression oracle returns a model that generalizes off-distribution. Finally, recent work on UCB with neural tangent kernels (Zhou et al., 2020) provides general regret bounds, but their bound is only sublinear under conditions such as the true reward function having small RKHS norm (see Remark 4.8 in their paper), which amounts to assuming they can recover the true parameters.

**Novelty**. We briefly discuss the novelty of our results compared to existing work. First, the results in Section 3 are novel. To the best of our knowledge, the proof strategy in our main result, Theorem 3.6, is novel, though we note that the preceding lemmas are based on standard arguments—e.g., bounding the convexity of $\tilde{L}_p(\phi)$ (Lemma 3.3) and the Lipschitz constant (Lemma 3.4) of $\tilde{f}_\phi$; also, Lemma 3.5 relies on a standard covering number argument. For our applications to bandits and transfer learning, our key novel results are Lemma C.3 for bandits, which proves smoothed bounded response for quadratic neural networks, and Lemma D.1 for transfer learning. Finally, to the best of our knowledge, our arguments in Section 6 are novel.

---

[3]Slightly different from realizability in PAC learning (Kearns et al., 1994), which says there is a model with zero true loss.

# B PROOFS FOR SECTION 3

## B.1 PROOF OF MINIMUM EIGENVALUE FOR UNIFORM DISTRIBUTION

In this section, we prove the claim that Assumption 3.2 holds for the covariate distribution where $x_i$ is an i.i.d. random variable with distribution $\text{Uniform}(x_i; [-1/2, 1/2])$. To this end, note that

$$
\begin{aligned}
\mathbb{E}_{p(x)}[(x^\top \Delta x)^2] &= \mathbb{E}_{p(x)}\left[\left(\sum_{i,j=1}^d x_i x_j \Delta_{ij}\right)^2\right] \\
&= \mathbb{E}_{p(x)}\left[\sum_i x_i^4 \Delta_{ii}^2 + \sum_{i \neq j} x_i^2 x_j^2 \Delta_{ii}\Delta_{jj} + 2\sum_{i \neq j} x_i^2 x_j^2 \Delta_{ij}^2\right] \\
&= \frac{1}{80}\sum_i \Delta_{ii}^2 + \frac{1}{144}\sum_{i \neq j}\Delta_{ii}\Delta_{jj} + \frac{1}{72}\sum_{i \neq j}\Delta_{ij}^2 \\
&= \left(\frac{1}{80} - \frac{1}{144}\right)\sum_i \Delta_{ii}^2 + \frac{1}{144}\left(\sum_i \Delta_{ii}\right)^2 + \frac{1}{72}\sum_{i \neq j}\Delta_{ij}^2 \\
&\geq \frac{1}{180}\|\Delta\|_F^2,
\end{aligned}
$$

as claimed. $\quad\square$

## B.2 PROOF OF LEMMA 3.3

We use the notation $U : \nabla^2 f(\phi) : V$ to denote the matrix inner product $\langle U, \nabla^2 f(\phi)(V)\rangle$ for $U, V \in \mathbb{R}^{d \times d}$. The Hessian $\nabla^2 f(\phi)$ can be viewed as a $d^2 \times d^2$ matrix. As everything here is bounded, we can exchange the expectation and differentiation. Therefore, the Hessian of our loss function has for any symmetric matrix $\Delta$

$$
\Delta : \nabla^2 \tilde{L}_p(\phi) : \Delta = 2\mathbb{E}_{p(x)}[(x^\top \Delta x)^2] \geq 2\alpha\|\Delta\|_F^2,
$$

where the last inequality uses Assumption 3.2. $\quad\square$

## B.3 PROOF OF LEMMA 3.4

By our definition, for any $\phi, \phi' \in \Phi$,
$$
|\tilde{f}_\phi(x) - \tilde{f}_{\phi'}(x)| = |(x^\top(\phi - \phi')x)^2| \leq x_{\max}^2\|\phi - \phi'\|_F.
$$
Given our quadratic loss function, we have
$$
\begin{aligned}
&|(\tilde{f}_\phi(x) - \tilde{f}_{\phi^*}(x))^2 - (\tilde{f}_{\phi'}(x) - \tilde{f}_{\phi^*}(x))^2| \\
&\leq |\tilde{f}_\phi(x) - \tilde{f}_{\phi^*}(x) + \tilde{f}_{\phi'}(x) - \tilde{f}_{\phi^*}(x)||\tilde{f}_\phi(x) - \tilde{f}_{\phi'}(x)| \\
&\leq 4\phi_{\max}x_{\max}^4\|\phi - \phi'\|_F.
\end{aligned}
$$
Next, the true loss satisfies
$$
|\tilde{L}_p(\phi) - \tilde{L}_p(\phi')| \leq \mathbb{E}_{p(x)}[|(\tilde{f}_\phi(x) - \tilde{f}_{\phi^*}(x))^2 - (\tilde{f}_{\phi'}(x) - \tilde{f}_{\phi^*}(x))^2|] \leq 4\phi_{\max}x_{\max}^4\|\phi - \phi'\|_F.
$$
Finally, the empirical loss satisfies
$$
\begin{aligned}
|\hat{\tilde{L}}(\phi; Z) - \hat{\tilde{L}}(\phi'; Z)| &= \left|\frac{1}{n}\sum_{i=1}^n[(\tilde{f}_\phi(x_i) - y_i)^2 - (\tilde{f}_{\phi'}(x_i) - y_i)^2]\right| \\
&\leq \frac{1}{n}\sum_{i=1}^n |(\tilde{f}_\phi(x_i) - \tilde{f}_{\phi^*}(x_i))^2 - (\tilde{f}_{\phi'}(x_i) - \tilde{f}_{\phi^*}(x_i))^2| \\
&\quad + \frac{1}{n}\sum_{i=1}^n |\xi_i| \cdot |\tilde{f}_\phi(x_i) - \tilde{f}_{\phi'}(x_i)| \\
&\leq (4\phi_{\max}x_{\max}^4 + 2\xi_{\max}x_{\max}^2)\|\phi - \phi'\|_F,
\end{aligned}
$$
as claimed. $\quad\square$

### B.4 PROOF OF LEMMA 3.5

First, we have the following results:

**Lemma B.1** (Covering Number of Euclidean Ball). *For a Euclidean ball in $\mathbb{R}^{n_1 \times n_2}$ with radius $R$ with respect to Frobenius norm, there exists an $\epsilon$-net $\mathcal{E}$ such that*

$$|\mathcal{E}| \leq \left(1 + \frac{2R}{\epsilon}\right)^{n_1 n_2}.$$

*Proof.* This claim follows by a direct application of Proposition 4.2.12 in Vershynin (2018). □

**Lemma B.2** (Hoeffding's Inequality for Subgaussian Random Variables). *Letting $\{z_i\}_{i=1}^n$ be a set of independent $\sigma$-subgaussian random variables, then for all $t \geq 0$, we have*

$$\Pr\left[\frac{1}{n}\sum_{i=1}^n z_i \geq t\right] \leq \exp\left(-\frac{2nt^2}{\sigma^2}\right).$$

*Proof.* See Proposition 2.1 of Wainwright (2016). □

Now, we prove Lemma 3.5. Consider an $\epsilon/(4K)$-net $\mathcal{E}$. Then, for any $\phi \in \Phi$, there exists $\phi' \in \mathcal{E}$ such that

$$|(\hat{\tilde{L}}(\phi; Z) - \tilde{L}_p(\phi)) - (\hat{\tilde{L}}(\phi'; Z) - \tilde{L}_p(\phi'))| \leq 2K\|\phi - \phi'\|_F \leq \frac{\epsilon}{2}.$$

Therefore, we have

$$\mathbb{P}_{p(Z)}\left[\sup_\theta |\hat{L}(g(\theta); Z) - L_p(g(\theta)) - \sigma(Z)| \geq \epsilon\right]$$

$$= \mathbb{P}_{p(Z)}\left[\sup_{\phi \in \Phi} |\hat{\tilde{L}}(\phi; Z) - \tilde{L}_p(\phi) - \sigma(Z)| \geq \epsilon\right]$$

$$\leq \mathbb{P}_{p(Z)}\left[\max_{\phi \in \mathcal{E}} |\hat{\tilde{L}}(\phi; Z) - \tilde{L}_p(\phi) - \sigma(Z)| \geq \frac{\epsilon}{2}\right]$$

$$\leq \sum_{\phi \in \mathcal{E}} \mathbb{P}_{p(Z)}\left[|\hat{\tilde{L}}(\phi; Z) - \tilde{L}_p(\phi) - \sigma(Z)| \geq \frac{\epsilon}{2}\right]. \tag{13}$$

Now, defining

$$\bar{\tilde{L}}(\phi; Z) = \frac{1}{n}\sum_{i=1}^n (\tilde{f}_\phi(x_i) - \tilde{f}_{\phi^*}(x_i))^2 \quad \text{and} \quad \tilde{\eta}(\phi; Z) = \frac{1}{n}\sum_{i=1}^n (\tilde{f}_\phi(x_i) - \tilde{f}_{\phi^*}(x_i))\xi_i,$$

and recalling that $\sigma(Z) = n^{-1}\sum_{i=1}^n \xi_i^2$, then we have

$$\hat{\tilde{L}}(\phi; Z) = \bar{\tilde{L}}(\phi; Z) + 2\tilde{\eta}(\phi; Z) + \sigma(Z).$$

Thus, continuing from (13), we have

$$\sum_{\phi \in \mathcal{E}} \mathbb{P}_{p(Z)}\left[|\hat{\tilde{L}}(\phi; Z) - \tilde{L}_p(\phi) - \sigma(Z)| \geq \frac{\epsilon}{2}\right]$$

$$\leq \sum_{\phi \in \mathcal{E}} \mathbb{P}_{p(Z)}\left[|\bar{\tilde{L}}(\phi; Z) - \tilde{L}_p(\phi)| + 2|\tilde{\eta}(\phi; Z)| \geq \frac{\epsilon}{2}\right]$$

$$\leq \sum_{\phi \in \mathcal{E}} \left(\mathbb{P}_{p(Z)}\left[|\bar{\tilde{L}}(\phi; Z) - \tilde{L}_p(\phi)| \geq \frac{\epsilon}{6}\right] + \mathbb{P}_{p(Z)}\left[|\tilde{\eta}(\phi; Z)| \geq \frac{\epsilon}{6}\right]\right). \tag{14}$$

For the first term in (14), note that $|(\tilde{f}_\phi(x) - \tilde{f}_{\phi^*}(x))^2| \leq \ell_{\max}^2$, so $(\tilde{f}_\phi(x) - \tilde{f}_{\phi^*}(x))^2$ is $\ell_{\max}^2$-subgaussian; thus, by Lemma B.2, we have

$$\sum_{\phi \in \mathcal{E}} \mathbb{P}_{p(Z)}\left[|\bar{\tilde{L}}(\phi; Z) - \tilde{L}_p(\phi)| \geq \frac{\epsilon}{6}\right] \leq 2|\mathcal{E}| \cdot \exp\left(-\frac{n\epsilon^2}{18\ell_{\max}^4}\right). \tag{15}$$

Next, for the second term in (14), note that $|(\tilde{f}_\phi(x_i) - \tilde{f}_{\phi^*}(x_i))\xi_i| \le \ell_{\max}\xi_{\max}$, so $(\tilde{f}_\phi(x_i) - \tilde{f}_{\phi^*}(x_i))\xi_i$ is $\ell_{\max}\xi_{\max}$-subgaussian; thus, by Lemma B.2, we have

$$\sum_{\phi \in \mathcal{E}} \mathbb{P}_{p(Z)}\left[|\tilde{\eta}(\phi; Z)| \ge \frac{\epsilon}{6}\right] \le 2|\mathcal{E}| \cdot \exp\left(-\frac{n\epsilon^2}{18\ell_{\max}^2\xi_{\max}^2}\right). \tag{16}$$

Combining (15) & (16), continuing from (14), we have

$$\sum_{\phi \in \mathcal{E}} \left(\mathbb{P}_{p(Z)}\left[|\tilde{\bar{L}}(\phi; Z) - \tilde{L}_p(\phi)| \ge \frac{\epsilon}{6}\right] + \mathbb{P}_{p(Z)}\left[|\tilde{\eta}(\phi; Z)| \ge \frac{\epsilon}{6}\right]\right)$$

$$\le 4|\mathcal{E}| \cdot \exp\left(-\frac{n\epsilon^2}{18\ell_{\max}^2(\ell_{\max}^2 + \xi_{\max}^2)}\right)$$

$$\le 2\left(1 + \frac{8\phi_{\max}K}{\epsilon}\right)^{d^2} \cdot \exp\left(-\frac{n\epsilon^2}{18\ell_{\max}^2(\ell_{\max}^2 + \xi_{\max}^2)}\right)$$

$$= 2\exp\left(-\frac{n\epsilon^2}{18\ell_{\max}^2(\ell_{\max}^2 + \xi_{\max}^2)} + d^2\log\left(1 + \frac{8\phi_{\max}K}{\epsilon}\right)\right), \tag{17}$$

where for the first inequality, we have used $\max\{\ell_{\max}^2, \xi_{\max}^2\} \le \ell_{\max}^2 + \xi_{\max}^2$, and the second inequality follows since by Lemma B.1, the covering number of the $\epsilon$-net $\mathcal{E}$ of $\Phi$ satisfies

$$|\mathcal{E}| \le \left(1 + \frac{2\phi_{\max}}{\epsilon}\right)^{d^2}.$$

Finally, we choose $\epsilon$ so that (17) is smaller than $\delta$—in particular, letting

$$\epsilon = \sqrt{\frac{18\ell_{\max}^2(\ell_{\max}^2 + \xi_{\max}^2)}{n}\left(d^2\max\left\{1, \log\left(1 + \frac{8\phi_{\max}Kn}{\ell_{\max}^2}\right)\right\} + \log\frac{2}{\delta}\right)}.$$

then continuing (17), we have

$$2\exp\left(-\frac{n\epsilon^2}{18\ell_{\max}^2(\ell_{\max}^2 + \xi_{\max}^2)} + d^2\log\left(1 + \frac{8\phi_{\max}K}{\epsilon}\right)\right) \le \delta,$$

as claimed. $\quad\square$

## B.5 Proof of Proposition 3.8

$\hat{\tilde{L}}(\phi; Z)$ is twice differentiable and convex in $\phi$. Note that the minimization problem of $\hat{L}(\theta; Z)$ is equivalent to that of $\hat{\tilde{L}}(g(\hat{\theta}); Z)$. We consider two cases. First, consider the case where $\hat{\theta}$ has rank $d$. The first order condition $\nabla\hat{L}(\theta; Z) = 0$ is the same as $\nabla\hat{\tilde{L}}(g(\hat{\theta}); Z) = 0$, which gives

$$\nabla\hat{\tilde{L}}(\hat{\phi}; Z)\hat{\theta} = 0. \tag{18}$$

As $\hat{\theta}$ is of full row rank, there exists a matrix $\hat{\theta}^\dagger \in \mathbb{R}^{k \times d}$ such that $\hat{\theta}\hat{\theta}^\dagger = I$ (e.g. $\hat{\theta}^\dagger = \hat{\theta}^\top(\hat{\theta}\hat{\theta}^\top)^{-1}$). We can right multiply the above equation by $\hat{\theta}^\dagger$ and obtain that

$$\nabla\hat{\tilde{L}}(\hat{\phi}; Z) = 0.$$

As $\hat{\tilde{L}}(\phi; Z)$ is convex in $\phi$, the above implies $\hat{\phi} = g(\hat{\theta})$ is a global minimum of $\hat{\tilde{L}}(\phi; Z)$. Therefore, $\hat{\theta}$ is a global minimum of $\hat{L}(\theta; Z)$. Next, consider the case where the rank of $\hat{\theta}$ is smaller than $d$. In this case, we follow the proof strategy in Proposition 4 in Bach et al. (2008); we provide here for completeness. In this case, Equation (18) still holds, which implies

$$0 = \nabla\hat{\tilde{L}}(\hat{\phi}; Z)\hat{\theta}\hat{\theta}^\top = \nabla\hat{\tilde{L}}(\hat{\phi}; Z)\hat{\phi}. \tag{19}$$

The Hessian of $\hat{\tilde{L}}(g(\hat{\theta}); Z)$ has

$$\nabla^2\hat{\tilde{L}}(g(\hat{\theta}); Z)(\Delta, \Delta) = 2\langle\nabla\hat{\tilde{L}}(\hat{\phi}; Z), \Delta\Delta^\top\rangle + \nabla^2\hat{\tilde{L}}(\hat{\phi}; Z)(\hat{\theta}\Delta^\top + \Delta\hat{\theta}^\top, \hat{\theta}\Delta^\top + \Delta\hat{\theta}^\top).$$

As $\hat{\theta}R$ is also a local minimum for any orthogonal matrix $R$ (i.e., $RR^\top = R^\top R = I$), we can find a $\hat{\theta}$ with the last column being 0 by right multiplying certain $R$. Then, consider any $\Delta$ with the first $k-1$ columns being 0 and the last column being any $u \in \mathbb{R}^d$. With this choice of $\Delta$ and $\hat{\theta}$, $\hat{\theta}\Delta^\top = 0$. Therefore,

$$\nabla^2 \hat{\tilde{L}}(g(\hat{\theta}); Z)(\Delta, \Delta) = 2u^\top \nabla \hat{\tilde{L}}(\hat{\phi}; Z)u.$$

Since $\hat{\theta}$ is a local minimum of $\hat{\tilde{L}}(g(\hat{\theta}); Z)$, it holds that $\nabla^2 \hat{\tilde{L}}(g(\hat{\theta}); Z)(\Delta, \Delta) \geq 0$, which implies

$$\nabla \hat{\tilde{L}}(\hat{\phi}; Z) \succeq 0. \tag{20}$$

Equation (19) and (20) together comprise the first order conditions of the convex minimization problem $\min_{\phi \succeq 0} \hat{\tilde{L}}(\phi; Z)$. Thus, $\hat{\theta}$ is also a global minimum. $\square$

## C  PROOFS FOR SECTION 4

First, we provide the full statement of Theorem 4.2 (including constants).

**Theorem C.1.** *The expected regret of Algorithm 1 is*

$$R(T) \leq C_0 + C_1 \cdot T^{2/3} \left( \log \left( 3 + \frac{8\phi_{max}KT}{\ell_{max}^2} \right) \right)^{1/3},$$

*where*

$$C_0 = \frac{64(\phi_{max})^{\frac{2d^2+2}{2d^2-1}}}{(135M(\ell_{max} + \xi_{max})^2 d^3)^{\frac{2d^2+2}{2d^2-1}} (8\phi_{max}K/\ell_{max}^2)^{\frac{3d^2}{2d^2-1}}},$$
$$C_1 = 162d^2(M^2(\ell_{max} + \xi_{max})^4 \phi_{max})^{1/3}.$$

Before we prove Theorem C.1, we first prove a preliminary result establishing an analog of the *smooth best arm response* property Rusmevichientong & Tsitsiklis (2010) to our setting. First, we have the following useful result:

**Lemma C.2.** *Let $\phi, \phi' \in \mathbb{R}^{d \times d}$ be symmetric matrices, let $x, x' \in \mathbb{R}^d$ be eigenvectors of $\phi, \phi'$ corresponding to their top eigenvalue, such that $\|x\|_2 = \|x'\|_2 = 1$, and let $\lambda_1 \geq \lambda_2 \geq ... \geq \lambda_d$ be the eigenvalues of $\phi'$. Suppose that $\langle x, x' \rangle \geq 0$. Then, we have*

$$\|x - x'\|_2 \leq \frac{2^{3/2}\|\phi - \phi'\|_2}{\lambda_1 - \lambda_2}.$$

*Proof.* See Corollary 3 of Yu et al. (2015). $\square$

Next, let $\chi : \mathbb{R}^{d \times d} \to 2^{\mathcal{X}}$ denote the subset of reward-maximizing arms for $g(\theta) = \theta\theta^\top$—i.e.,

$$\chi(\phi) = \arg\max_{x \in \mathcal{X}} x^\top \phi x,$$

where the argmax returns the set of all optimal values. Then, we have the following analog of smooth best arm response:

**Lemma C.3.** *For any $\phi \in \mathbb{R}^{d \times d}$, there exists $x \in \chi(\phi)$ and $x^* \in \chi(\phi^*)$ such that*
$$\|x - x^*\|_2 \leq M\|\phi - \phi^*\|_F.$$

*Proof.* First, note that $x, x^*$ are eigenvectors of $\phi, \phi^*$ corresponding to their top eigenvalues, respectively. Next, note that if $x^* \in \chi(\phi^*)$, then we also have $-x^* \in \chi(\phi^*)$; thus, without loss of generality, we can assume that $\langle x^*, x \rangle \geq 0$. Also, note that $\|x\|_2 = \|x^*\|_2 = 1$ since the optimizer maximizes the magnitude of $x$. Thus, we have

$$\|x - x^*\|_2 \leq \frac{2^{3/2}\|\phi - \phi^*\|_2}{\lambda_1 - \lambda_2} \leq M\|\phi - \phi^*\|_F,$$

where the second inequality follows by by Lemma C.2, and the third inequality follows by Assumption 4.1, as claimed. $\square$

Now, we prove Theorem C.1. The cumulative regret $R(T)$ of a horizon of $T$ has that

$$
\begin{aligned}
R(T) &= \mathbb{E}\left[\sum_{t=1}^{T}(f_{\theta^*}(x^*) - f_{\theta^*}(x_t))\right] \\
&= \mathbb{E}\left[\sum_{t=1}^{m}(f_{\theta^*}(x^*) - f_{\theta^*}(x_t)) + \sum_{t=m+1}^{T}(f_{\theta^*}(x^*) - f_{\theta^*}(x_t))\right] \\
&\leq 2m\phi_{\max} + \mathbb{E}\left[\sum_{t=m+1}^{T}\langle g(\hat\theta) - g(\theta^*), \hat x\hat x^\top - x^*x^{*\top}\rangle + \sum_{t=m+1}^{T}\langle g(\hat\theta), x^*x^{*\top} - \hat x\hat x^\top\rangle\right],
\end{aligned}
\tag{21}
$$

where $\hat\theta$ is an estimator that minimizes the empirical loss of the first $m$ samples, $\hat x \in \chi(g(\hat\theta))$ maximizes the estimated expected reward $f_{\hat\theta}(x)$, and $\langle \phi, \phi'\rangle = \sum_{i,j=1}^{d}\phi_{ij}\phi'_{ij}$ is the matrix inner product. Since $\hat x$ is a maximizer of $f_{\hat\theta}(x) = \langle g(\hat\theta), xx^\top\rangle$, we have $\langle g(\hat\theta), x^*x^{*\top} - \hat x\hat x^\top\rangle \leq 0$. Thus, continuing from (21), we have

$$
\begin{aligned}
R(T) &\leq 2m\phi_{\max} + \mathbb{E}\left[\sum_{t=m+1}^{T}\langle g(\hat\theta) - g(\theta^*), \hat x\hat x^\top - x^*x^{*\top}\rangle\right] \\
&\leq 2m\phi_{\max} + (T-m)\mathbb{E}\left[\|g(\hat\theta) - g(\theta^*)\|_F\|\hat x\hat x^\top - x^*x^{*\top}\|_F\right].
\end{aligned}
\tag{22}
$$

To bound the second term in (22), note that

$$
\|\hat x\hat x^\top - x^*x^{*\top}\|_F \leq \|\hat x\hat x^\top - \hat x x^{*\top}\|_F + \|\hat x x^{*\top} - x^*x^{*\top}\|_F \leq 2M\|g(\hat\theta) - g(\theta^*)\|_F,
$$

where the last step follows by Lemma C.3. Next, by Theorem 3.6, we have

$$
\begin{aligned}
\|g(\hat\theta) - g(\theta^*)\|_F &\leq \sqrt{\frac{2\epsilon}{\alpha}} \\
&= d\left(\frac{45^3\ell_{\max}^2(\ell_{\max}^2 + \xi_{\max}^2)}{10m}\left(d^2\max\left\{1, \log\left(1 + \frac{8\phi_{\max}Km}{\ell_{\max}^2}\right)\right\} + \log\frac{2}{\delta}\right)\right)^{1/4}
\end{aligned}
$$

with probability at least $1 - \delta$. Now, defining the event

$$
\mathcal{G} = \left\{\|g(\hat\theta) - g(\theta^*)\|_F \leq \sqrt{\frac{2\epsilon}{\alpha}}\right\},
$$

letting

$$
\delta = 2\exp\left(-d^2\max\left\{1, \log\left(1 + \frac{8\phi_{\max}Km}{\ell_{\max}^2}\right)\right\}\right),
$$

and continuing from (22), we have

$$
\begin{aligned}
R(T) &\leq 2m\phi_{\max} + T\cdot\mathbb{E}\left[\|g(\hat\theta) - g(\theta^*)\|_F\|\hat x\hat x^\top - x^*x^{*\top}\|_F\mathbb{1}(\mathcal{G})\right] + 4T\phi_{\max}\cdot\mathbb{P}(\mathcal{G}^c) \\
&\leq 2m\phi_{\max} + 2MT\cdot\mathbb{E}\left[\|g(\hat\theta) - g(\theta^*)\|_F^2\ \big|\ \mathcal{G}\right] + 4T\phi_{\max}\cdot\mathbb{P}(\mathcal{G}^c) \\
&\leq 2m\phi_{\max} + 270M(\ell_{\max} + \xi_{\max})^2 d^3 T\sqrt{\frac{\log(3 + 8\phi_{\max}KT/\ell_{\max}^2)}{m}} + \frac{8T\phi_{\max}}{(8\phi_{\max}Km/\ell_{\max}^2)^{d^2}}.
\end{aligned}
\tag{23}
$$

The third term in inequality (23) is smaller than the second term when

$$
m \geq \left(\frac{4\phi_{\max}}{135M(\ell_{\max} + \xi_{\max})^2 d^3(8\phi_{\max}K/\ell_{\max}^2)^{d^2}\sqrt{\log(3 + 8\phi_{\max}KT/\ell_{\max}^2)}}\right)^{1/(d^2-1/2)}.
\tag{24}
$$

For a choice of $m$ satisfying (24), continuing from (23), we have

$$R(T) \leq 2m\phi_{\max} + 540M(\ell_{\max} + \xi_{\max})^2 d^3 T \sqrt{\frac{\log(3 + 8\phi_{\max}KT/\ell_{\max}^2)}{m}}. \tag{25}$$

Next, we choose $m$ to minimize the upper bound in (24) for sufficiently large $T$—in particular,

$$m = \left\lceil \left( \frac{135M(\ell_{\max} + \xi_{\max})^2 d^3 T \sqrt{\log(3 + 8\phi_{\max}KT/\ell_{\max}^2)}}{\phi_{\max}} \right)^{\frac{2}{3}} \right\rceil. \tag{26}$$

With this choice of $m$, we have

$$R(T) \leq 162(M^2(\ell_{\max} + \xi_{\max})^4 \phi_{\max})^{\frac{1}{3}} d^2 T^{2/3} (\log(3 + 8\phi_{\max}KT/\ell_{\max}^2))^{\frac{1}{3}}.$$

Finally, note that (24) holds under the choice of $m$ in (26) for $T$ satisfying

$$T\sqrt{\log(3 + 8\phi_{\max}KT/\ell_{\max}^2)} \geq \frac{64(\phi_{\max})^{\frac{2d^2+2}{2d^2-1}}}{(135M(\ell_{\max} + \xi_{\max})^2 d^3)^{\frac{2d^2+2}{2d^2-1}} (8\phi_{\max}K/\ell_{\max}^2)^{\frac{3d^2}{2d^2-1}}}.$$

The claim follows. $\quad\square$

## D   PROOF OF THEOREM 5.2

First, we have the following key result:

**Lemma D.1.** *Let $\theta, \theta' \in \mathbb{R}^{d \times k}$, and let $\phi = \theta\theta^\top$ and $\phi' = \theta'\theta'^\top$. Assume that $\|\phi - \phi'\|_F \leq \eta$, and that $\sigma_{min}(\theta) \geq \sigma_0 > 0$, where $\sigma_{min}(\theta)$ is the minimum singular value of $\theta$ (more precisely, the dth largest singular value). Then, there exist orthogonal matrices $R, R' \in \mathbb{R}^{k \times k}$ such that*

$$\|\theta R - \theta'R'\|_F \leq \frac{\eta}{\sigma_0}. \tag{27}$$

*Proof.* Consider the SVDs $\theta = U\Sigma V^\top$ and $\theta' = U'\Sigma'V'^\top$, where $U, U' \in \mathbb{R}^{d \times d}$, $\Sigma, \Sigma' \in \mathbb{R}^{d \times d}$, and $V, V' \in \mathbb{R}^{k \times d}$; then, we have $\phi = U\Sigma^2 U^\top$ and $\phi' = U'\Sigma'^2 U'^\top$. Then, we claim that the choices $R = VU^\top$ and $R' = V'U'^\top$ satisfy (27). In particular, note that $\theta R = U\Sigma U^\top$ and $\theta'R' = U'\Sigma'U'^\top$, since $V^\top V = V'^\top V' = I_d$ since $k \geq d$, where $I_d \in \mathbb{R}^{d \times d}$ is the $d$-dimensional identity matrix. Thus, it suffices to show that

$$\sigma_0\|U\Sigma U^\top - U'\Sigma'U'^\top\|_F \leq \eta. \tag{28}$$

To this end, note that

$$\eta \geq \|\phi - \phi'\|_F = \|U\Sigma^2 U^\top - U'\Sigma'^2 U'^\top\|_F = \|U'^\top U\Sigma^2 - \Sigma'^2 U'^\top U\|_F, \tag{29}$$

where in the last step, we have multiplied the expression inside the Frobenius norm by $U'^\top$ on the left and by $U$ on the right, using the fact that the Frobenius norm is invariant under multiplication by orthogonal matrices. Defining $W = U'^\top U$, note that

$$(W\Sigma)_{ij} = \sum_{k=1}^d W_{ik}\Sigma_{kj} = W_{ij}\Sigma_{jj} \tag{30}$$

$$(\Sigma W)_{ij} = \sum_{k=1}^d \Sigma'_{ik}W_{kj} = W_{ij}\Sigma'_{ii} \tag{31}$$

$$(W\Sigma^2)_{ij} = \sum_{k=1}^d W_{ik}(\Sigma^2)_{kj} = W_{ij}\Sigma_{jj}^2 \tag{32}$$

$$(\Sigma^2 W)_{ij} = \sum_{k=1}^d (\Sigma'^2)_{ik}W_{kj} = W_{ij}\Sigma_{ii}'^2. \tag{33}$$

Then, continuing from (29), we have

$$\eta^2 \geq \|W\Sigma^2 - \Sigma'^2 W\|_F^2 = \sum_{i,j=1}^{d} W_{ij}^2 (\Sigma_{jj}^2 - \Sigma_{ii}'^2)^2$$

$$= \sum_{i,j=1}^{d} W_{ij}^2 (\Sigma_{jj} - \Sigma_{ii}')^2 (\Sigma_{jj} + \Sigma_{ii}')^2$$

$$\geq \sum_{i,j=1}^{d} W_{ij}^2 (\Sigma_{jj} - \Sigma_{ii}')^2 \sigma_0^2$$

$$= \sigma_0^2 \|W\Sigma - \Sigma'W\|_F^2$$

$$= \sigma_0^2 \|U'^\top U\Sigma - \Sigma'U'^\top U\|_F^2$$

$$= \sigma_0^2 \|U\Sigma U^\top - U'\Sigma'U'^\top\|_F^2,$$

where on the first line, we have used (32) & (33), on the third line we have used $\Sigma_{jj} \geq \sigma_0$, on the fourth line we have used (30) & (31), and on the last line we have multiplied on by $U'$ on the left $U^\top$ on the right, again using the fact that the Frobenius norm is invariant under multiplication by orthogonal matrices. Thus, we have shown (28), so the claim follows. $\qquad\square$

We note here that our result provides an analog of Lemma 6 in Ge et al. (2017a) for quadratic neural networks.

Now, we prove Theorem 5.2. First, by directly applying the arguments in the proof of Theorem 3.6, we have

$$\|g(\hat{\theta}_p) - g(\theta_p^*)\|_F \leq \sqrt{\frac{2\epsilon_p}{\alpha}}$$

with probability at least $1 - \delta/2$. However, $\hat{\theta}_p$ itself may not be close to $\theta_p^*$. Instead, applying Lemma D.1 with $\theta = \hat{\theta}_p$ and $\theta' = \theta_p^*$, and with $\eta = \sqrt{2\epsilon_p/\alpha}$, there exists an orthogonal matrix $R_p = R'R^\top$ that "aligns" $\hat{\theta}_p$ with $\theta_p^*$, yielding

$$\|\hat{\theta}_p - \theta_p^* R_p\|_F \leq \frac{1}{\sigma_0}\sqrt{\frac{2\epsilon_p}{\alpha}},$$

where $\sigma_0$ is the minimum singular value of $\theta_p^*$. Now, let $\tilde{\theta}_g = \theta_g^* R_p$, and note that this is a global minimizer (i.e., $g(\tilde{\theta}_g) = g(\theta_g^*)$), since $R_p$ is orthogonal. Then, we have

$$\|\tilde{\theta}_g - \hat{\theta}_p\|_F \leq \|\theta_g^* R_p - \theta_p^* R_p\|_F + \|\theta_p^* R_p - \hat{\theta}_p\|_F$$

$$\leq \|\theta_g^* - \theta_p^*\|_F + \frac{1}{\sigma_0}\sqrt{\frac{2\epsilon_p}{\alpha}}$$

$$\leq B + \frac{1}{\sigma_0}\sqrt{\frac{2\epsilon_p}{\alpha}} \tag{34}$$

with probability at least $1 - \delta/2$. In other words, an alternative global minimizer $\tilde{\theta}_g$ exists within a small Frobenius norm of our proxy estimator $\hat{\theta}_p$, even if $\hat{\theta}_p$ is not close to $\theta_p^*$.

Finally, on the event that (34) holds, note that for $\theta \in B_2(\hat{\theta}_p, \hat{B})$, we have the alternative upper bound

$$|f_\theta(x) - f_{\theta_g^*}(x))| \leq K\|g(\theta) - g(\theta_g^*)\|_F \leq K\hat{B},$$

where the first inequality holds by Lemma 3.4; thus, we can take $\ell_{\max} = K\hat{B}$. Thus, on the event that (34) holds, by Theorem 3.6, we have

$$\mathbb{P}_{p(Z)}\left[L_q(\hat{\theta}_g) \leq \frac{2K^2\epsilon_g}{\alpha}\right] \geq 1 - \frac{\delta}{2},$$

so the claim follows. $\quad\square$

# E    PROOFS FOR SECTION 6

## E.1    PROOF OF PROPOSITION 6.4

Suppose that $z_t \in \{0, 1\}$ is binary, $z_0 = 0$, and

$$\tilde{p}(z \mid z') = \begin{cases} 1 & \text{if } z = z' \\ 0 & \text{otherwise.} \end{cases}$$

In particular, since $z_0 = 0$, $p(w) = \mathbb{1}(w = w_0)$ places all weight on the zero sequence $w_0 = 0...0$. Next, consider the shifted distribution

$$\tilde{q}(z_t \mid z_{t-1}) = \begin{cases} 1 & \text{if } z = z' = 1 \\ 1 - \alpha/2 & \text{if } z = z' = 0 \\ \alpha/2 & \text{otherwise.} \end{cases}$$

Note that $\|\tilde{p}(\cdot \mid z') - \tilde{q}(\cdot \mid z')\|_{\text{TV}} \le \alpha$, so Assumption 6.3 is satisfied. Note that

$$q(w_0) = \prod_{t=1}^{T} \tilde{q}(0 \mid 0) = (1 - \alpha/2)^T.$$

As a consequence, we have

$$\begin{aligned} \|p - q\|_{\text{TV}} &= \sum_{w \in \mathcal{W}} |p(w) - q(w)| \\ &= |p(w_0) - q(w_0)| + \sum_{w \in \mathcal{W} \setminus \{w_0\}} q(w) \\ &= (1 - (1 - \alpha/2)^T) + (1 - (1 - \alpha/2)^T) \\ &= 2(1 - (1 - \alpha/2)^T), \end{aligned}$$

as claimed.    $\square$

## E.2    PROOF OF LEMMA 6.5

Note that

$$\begin{aligned} &\|\tilde{q}_t - \tilde{p}_t\|_{\text{TV}} \\ &= \sum_{j=1}^{k} \int |\tilde{q}_t(z, j) - \tilde{p}_t(z, j)| dz \\ &= \sum_{j=1}^{k} \sum_{j'=1}^{k} \int \mathbb{1}(j = \tilde{g}^*(z', j')) \cdot |\tilde{q}(z \mid z')\tilde{q}_{t-1}(z', j') - \tilde{p}(z \mid z')\tilde{p}_{t-1}(z', j')| dz' dz \\ &= \sum_{j'=1}^{k} \int |\tilde{q}(z \mid z')\tilde{q}_{t-1}(z', j') - \tilde{p}(z \mid z')\tilde{p}_{t-1}(z', j')| dz' dz \\ &\le \sum_{j'=1}^{k} \int |\tilde{q}(z \mid z') - \tilde{p}(z \mid z')| \cdot \tilde{q}_{t-1}(z', j') + \tilde{p}(z \mid z') \cdot |\tilde{q}_{t-1}(z', j') - \tilde{p}_{t-1}(z', j')| dz' dz \\ &\le \sum_{j'=1}^{k} \int \alpha \cdot \tilde{q}_{t-1}(z', j') + |\tilde{q}_{t-1}(z', j') - \tilde{p}_{t-1}(z', j')| dz' \\ &\le \alpha + \|\tilde{q}_{t-1} - \tilde{p}_{t-1}\|_{\text{TV}}. \end{aligned}$$

Since $q_0(z, j) = p_0(z, j)$ for all $z \in \mathcal{Z}$ and $j \in [k]$, by induction, $\|\tilde{q}_t - \tilde{p}_t\|_{\text{TV}} \le t\alpha$. Thus, we have

$$\|\tilde{q} - \tilde{p}\|_{\text{TV}} \le \frac{1}{T} \sum_{t=1}^{T} \|\tilde{q}_t - \tilde{p}_t\| \le T\alpha,$$

as claimed.    $\square$

### E.3 PROOF OF LEMMA 6.6

First, we prove the following lemma.

**Lemma E.1.** *We have*

$$\tilde{p}_t(z_t, j_{t-1}) = \sum_{j_1,...,j_{t-2}} \int \left( \prod_{\tau=1}^{t-1} \mathbb{1}(j_\tau = \tilde{g}^*(z_\tau, j_{\tau-1})) \right) \cdot p(z_1, ..., z_t) dz_1...dz_{t-1}.$$

*Proof.* For the base case, we have

$$\tilde{p}_2(z_2, j_1) = \sum_{j_0=1}^{k} \int \mathbb{1}(j_1 = \tilde{g}^*(z_1, j_0)) \cdot \tilde{p}(z_2 \mid z_1) \cdot \tilde{p}_1(z_1, j_0) dz_1$$

$$= \sum_{j_0=1}^{k} \int \mathbb{1}(j_1 = \tilde{g}^*(z_1, j_0)) \cdot \tilde{p}(z_2 \mid z_1) \cdot \mathbb{1}(j = 0) \cdot \tilde{p}(z_1) dz_1$$

$$= \int \mathbb{1}(j_1 = \tilde{g}^*(z_1, j_0)) \cdot p(z_1, z_2) dz_1,$$

as claimed. For the inductive case, we have

$$\tilde{p}_t(z_t, j_{t-1}) = \sum_{j_{t-2}=1}^{k} \int \mathbb{1}(j_{t-1} = \tilde{g}^*(z_{t-1}, j_{t-2})) \cdot \tilde{p}(z_t \mid z_{t-1}) \cdot \tilde{p}_{t-1}(z_{t-1}, j_{t-2}) dz_{t-1}$$

$$= \sum_{j_1,...,j_{t-2}=1}^{k} \int \left( \prod_{\tau=1}^{t-1} \mathbb{1}(j_{\tau-1} = \tilde{g}^*(z_{\tau-1}, j_{\tau-2})) \right) \cdot p(z_1, ..., z_t) dz_1...dz_{t-1}.$$

as claimed. □

Now, we prove Lemma 6.6. First, note that for each $t \in [T]$, we have

$$\mathbb{P}_{p(w)} \left[ (\hat{g}(w)_t \neq g^*(w)_t) \wedge \left( \bigwedge_{\tau=1}^{t-1} \hat{g}(w)_\tau = g^*(w)_\tau \right) \right]$$

$$= \int \mathbb{1}(\hat{g}(w)_t \neq g^*(w)_t) \cdot \left( \prod_{\tau=1}^{t-1} \mathbb{1}(\hat{g}(w)_\tau = g^*(w)_\tau) \right) \cdot p(w) dw$$

$$= \sum_{j_1,...,j_{t-1}=1}^{k} \int \mathbb{1}(\hat{g}(w)_t \neq g^*(w)_t) \cdot \left( \prod_{\tau=1}^{t-1} \mathbb{1}(\hat{g}(w)_\tau = g^*(w)_\tau) \right) \cdot p(j_1...j_{t-1} \mid w) \cdot p(w) dw$$

$$= \sum_{j_1,...,j_{t-1}=1}^{k} \int \mathbb{1}(\hat{g}(w)_t \neq g^*(w)_t) \cdot \cdot \left( \prod_{\tau=1}^{t-1} \mathbb{1}(\hat{g}(w)_\tau = g^*(w)_\tau) \right) \cdot \left( \prod_{\tau=1}^{t-1} \mathbb{1}(j_\tau = \tilde{g}^*(z_\tau, j_{\tau-1})) \right)$$

$$\cdot p(w) dw$$

$$= \sum_{j_1,...,j_{t-1}=1}^{k} \int \mathbb{1}(\hat{\tilde{g}}(z_\tau, j_{\tau-1}) \neq \tilde{g}^*(z_\tau, j_{\tau-1})) \cdot \left( \prod_{\tau=1}^{t-1} \mathbb{1}(\hat{\tilde{g}}(z_\tau, j_{\tau-1}) = \tilde{g}^*(z_\tau, j_{\tau-1})) \right)$$

$$\cdot \left( \prod_{\tau=1}^{t-1} \mathbb{1}(j_\tau = \tilde{g}^*(z_\tau, j_{\tau-1})) \right) \cdot p(w) dw$$

$$\leq \sum_{j_1,...,j_{t-1}=1}^{k} \int \mathbb{1}(\hat{\tilde{g}}(z_\tau, j_{\tau-1}) \neq \tilde{g}^*(z_\tau, j_{\tau-1})) \cdot \left( \prod_{\tau=1}^{t-1} \mathbb{1}(j_\tau = \tilde{g}^*(z_\tau, j_{\tau-1})) \right) \cdot p(w) dw$$

$$= \sum_{j_1,...,j_{t-1}=1}^{k} \int \mathbb{1}(\hat{\tilde{g}}(z_\tau, j_{\tau-1}) \neq \tilde{g}^*(z_\tau, j_{\tau-1})) \cdot \left( \prod_{\tau=1}^{t-1} \mathbb{1}(j_\tau = \tilde{g}^*(z_\tau, j_{\tau-1})) \right) \cdot p(z_1, ..., z_t) dz_1...dz_t$$

$$= \mathbb{P}_{p_t(z,j)} \left[ \hat{\tilde{g}}(z_\tau, j_{\tau-1}) \neq \tilde{g}^*(z_\tau, j_{\tau-1}) \right],$$

where the last step follows from Lemma E.1. Now, note that

$$\mathbb{P}_{p(w)}[\hat{g}(w) \neq g^*(w)] = \sum_{t=1}^{T} \mathbb{P}_{p(w)}\left[(\hat{g}(w)_t \neq g^*(w)_t) \wedge \left(\bigwedge_{\tau=1}^{t-1} \hat{g}(w)_\tau = g^*(w)_\tau\right)\right]$$

$$\leq \sum_{t=1}^{T} \mathbb{P}_{\tilde{p}_t(z,j)}\left[\hat{\tilde{g}}(z,j) \neq \tilde{g}^*(z,j)\right]$$

$$\leq T\epsilon_g,$$

as claimed.  $\square$

### E.4  PROOF OF THEOREM 6.7

First, we show that $\mathbb{P}_{q(z,j)}[\hat{\tilde{g}}(z,j) \neq \tilde{g}^*(z,j)] \leq \epsilon_g + T\alpha$. To this end, note that

$$\mathbb{P}_{q(z,j)}[\hat{\tilde{g}}(z,j) \neq \tilde{g}^*(z,j)]$$

$$= \mathbb{P}_{p(z,j)}[\hat{\tilde{g}}(z,j) \neq \tilde{g}^*(z,j)] + \mathbb{P}_{q(z,j)}[\hat{\tilde{g}}(z,j) \neq \tilde{g}^*(z,j)] - \mathbb{P}_{p(z,j)}[\hat{\tilde{g}}(z,j) \neq \tilde{g}^*(z,j)]$$

$$\leq \epsilon_g + \sum_{j=1}^{k} \int \mathbb{1}(\hat{\tilde{g}}(z,j) \neq \tilde{g}^*(z,j)) \cdot |\tilde{q}(z,j) - \tilde{p}(z,j)| dz$$

$$\leq \epsilon_g + \|\tilde{q} - \tilde{p}\|_{\text{TV}}$$

$$\leq \epsilon_g + T\alpha.$$

Next, by Lemma 6.6 with $q$ in place of $p$ and $\epsilon_g + T\alpha$ in place of $\epsilon_g$, we have $\mathbb{P}_{q(w)}[\hat{g}(w) \neq g^*(w)] \leq T\epsilon_g + T^2\alpha$. Then, assuming that $\hat{g}(w) = g^*(w)$, we have

$$\|f^*(x,w) - \hat{f}(x,w)\|_2$$

$$= \|(f^*_{j_T} \circ ... \circ f^*_{j_1})(x) - (\hat{f}_{j_T} \circ ... \circ \hat{f}_{j_1})(x)\|_2$$

$$\leq \sum_{t=1}^{T} \|(f^*_{j_T} \circ ... \circ f^*_{j_{t+1}} \circ f^*_{j_t} \circ \hat{f}_{j_{t-1}} \circ ... \circ \hat{f}_{j_1})(x) - (f^*_{j_T} \circ ... \circ f^*_{j_{t+1}} \circ \hat{f}_{j_t} \circ \hat{f}_{j_{t-1}} \circ ... \circ \hat{f}_{j_1})(x)\|_2$$

$$\leq \sum_{t=1}^{T} K^{T-t} \cdot \|(f^*_{j_t} \circ \hat{f}_{j_{t-1}} \circ ... \circ \hat{f}_{j_1})(x) - (\hat{f}_{j_t} \circ \hat{f}_{j_{t-1}} \circ ... \circ \hat{f}_{j_1})(x)\|_2$$

$$\leq \sum_{t=1}^{T} K^{T-t}\epsilon_f$$

$$\leq T\epsilon_f \cdot \max\{K^{T-1}, 1\}.$$

The claim follows by a union bound.  $\square$

