# OpenReview forum: "Robust Generalization of Quadratic Neural Networks via Function Identification"
_ICLR.cc/2022/Conference — ICLR 2022 Submitted_

### Official Review · Reviewer_QEKW · 2021-10-26

**Correctness:** 3
**Technical Novelty And Significance:** 2
**Empirical Novelty And Significance:** 1
**Recommendation:** 5
**Confidence:** 3

**Main Review:**

Strengths: I think the technical extension and its applications presented in the paper are conceptually interesting

Weaknesses:
1. The technical details of the paper (at least the main paper; I don't have access to the Appendix yet) is not very satisfactory: here are some examples
 - Lemma 6.2, the statement contains G (Rademacher complexity of G), but G is never defined
 - In section 6, it seems to me that the neural networks in neural network modules can only be quadratic NN (in order to apply Theorem 3.6). How realistic is this? And the authors should also state this fact in the paper
 - In Lemma 6.5, \tilde q and \tilde p are used in the statement and the discussion above, but were never defined; these definitions are important for the reader to understand the seemingly contradictory result of Proposition 6.4 and Lemma 6.5.

2. The assumptions made in the paper seem to be a little too strong to me:
 - It assumes that the noise are independent of the data, which doesn't seem to be very common in robust learning; or in rebuttal, perhaps the authors should discuss/cite related work to justify this assumption
 - The single-layer quadratic NN seems to me a little over-simplified to be studied; or the authors should provide clear argument on why this is not the case

3. The difficulty of the technical extension is not very obvious to me in the current paper: either the authors should clearly discuss and highlight these difficulties in the main paper or they should try to strengthen their technical result in a future version


**Summary Of The Paper:**

Under the generative data paradigm (e.g., the hypothesis space contains the ground-truth), the paper extends bounds on test (generalization) error under data distribution shift from the common setup when parameter identification is assumed to the setup where the ground-truth parameter may not be identifiable (not unique). Then the paper shows how this technical extension can be applied to obtain test-error bounds under distributional shift in three scenarios: bandits represented by an over-parametrized quadratic NN; transfer learning with both covariate and label shift; neural network modules

**Summary Of The Review:**

While I think the paper works on an interesting topic with several interesting applications, I feel technical quality of the current version of the paper lacking strength; however, I would not strongly argue for rejection if the authors can provide convincing rebuttal or are willing to improve on the technical quality of the paper.

---

> ### Author Response · Authors · 2021-11-23
> **Response to Reviewer QEKW**
>
> Thank you for your helpful comments and feedback!
>
> **Technical details.**
> Thank you for pointing out these issues; we have fixed them in our paper. We note that in Section 6, we assume the components $f_j$ are quadratic neural networks. However, this assumption is only needed to prove Lemma 6.1; in general, our subsequent results (in particular, Theorem 6.7) hold for any model that satisfies the statement of Lemma 6.1. Finally, we have included pointers to definitions of $\tilde{p}$ and $\tilde{q}$ in our paper; importantly, Proposition 6.4 refers to $p$ and $q$, whereas Lemma 6.5 refers to $\tilde{p}$ and $\tilde{q}$.
>
> **Noise independence.**
> We are considering a different kind of robustness---namely, robustness to distribution shift. This challenge has attracted much attention in the recent deep learning literature---e.g., [2] introduced the ImageNet-C dataset of synthetic distribution shifts, and [3] introduced the Wilds dataset, both to study this problem. Assuming independent noise is reasonable in this problem setting. To the best of our knowledge, this problem has received little (if any) attention from the theory community.
>
> **Single-layer QNN.** We agree that studying other activation functions or deeper architectures is an important next step, but believe these are beyond the scope of our work. Importantly, on-distribution generalization of neural networks was first shown for simple models such as quadratic neural networks [1], before being extended to more complex architectures; we believe robust generalization will follow a similar trend. We will add a discussion of future work along these lines.
>
> **Technical difficulty.**
> First, we note that our main contribution is to explain a phenomenon (namely, robust generalization for overparameterized models). In particular, to the best our knowledge, this problem has not been studied in the deep learning theory literature, and all of our theorems are novel. Finally, we have added a detailed discussion of technical novelty to Appendix A.
>
> [1] Du, Simon, and Jason Lee. "On the power of over-parametrization in neural networks with quadratic activation." International Conference on Machine Learning. PMLR, 2018.
>
> [2] Hendrycks, Dan, and Thomas Dietterich. "Benchmarking neural network robustness to common corruptions and perturbations." ICLR, 2019.
>
> [3] Koh, Pang Wei, et al. "Wilds: A benchmark of in-thewild distribution shifts." arXiv preprint arXiv:2012.07421 2 (2020).

---

### Official Review · Reviewer_LNcd · 2021-11-02

**Correctness:** 4
**Technical Novelty And Significance:** 3
**Empirical Novelty And Significance:** 3
**Recommendation:** 5
**Confidence:** 4

**Main Review:**

The paper investigates a very relevant and interesting topic of uncertainty estimation in overparameterized neural networks (here with one hidden layer network with quadratic activation function). The interesting and novel part of the paper is in attempt to give uncertainty estimation for predictors rather than parameters. The later is of course problematic in neural networks, since equivalent predictors can have different parameters (as acknowledged by the paper).
The paper starts of by bounding an excess risk through a fairly standard covering number argument. At this point the idea is to "convert" excess risk to a gap between predictor and the regression function. Perhaps, here comes the novel idea, namely observing that the risk is $\\alpha$-strongly convex in parameters of a neural network, and that the predictor is Lipschitz. Here strong convexity heavily relies on the quadratic nature of a neural network (one can see that by computing the Hessian). Interestingly, for considered covariate instance, $\\alpha = \\text{poly}(d)$ (where $d$ is a covariate dimension).

The resulting estimation bound is of order $\\text{poly}(d B) / (\\alpha \\sqrt{n})$, where the paper assumes that $\\|\theta\\|_F, \\|\theta^{\\star}\\|_F \\leq B$ such that $\\theta$, $\\theta^{\\star}$ are parameters of a fitted network and that of the regression function network.
While dependence on $\\alpha$ (somewhat surprisingly) is not problematic, boundedness of $\\|\theta\\|_F$ seems to be a bit problematic. When fitting overparameterized networks (especially till interpolation), $\\|\theta\\|_F^2$ can scale linearly in $n$, which will make the bound vacuous. This is of course a pessimistic view, but to have an entire picture, we would need to have an algorithm-dependent bound on the norm.

This limitation later finds its way into the explore-and-commit-type algorithm, where one commits to the argmin predictor (which is an interpolant in the overparameterized case). Here, the length of the exploration stage $m$ is tuned based on $d, B$ among other parameters.
The resulting regret depends on the norm through constants $C_0$, $C_1$ (at this point I wouldn't call them constants). Can we ensure that they don't scale as $\\text{poly}(m)$?

It is likely that this issue can be mitigated by analyzing the norm (e.g. given by GD solution), and for instance using some arguments as in Arora et al. 2019, however I feel that right now the entire picture is not very clear.

On another note, what are the challenges in analyzing a UCB-style algorithm employing the confidence bound proved here?

**Summary Of The Paper:**

The paper considers uncertainty estimation in overparameterized shallow neural networks with quadratic activation function. In particular, the paper assumes a non-linear regression model where labels are generated by an aforementioned neural net f* and noise is bounded. Then, uncertainty estimation is in giving a confidence interval around f*(x) for any given x. This differs from the usual uncertainty estimation in the linear regression, where one is concerned with deviation of parameters. The bound is a combination of the uniform convergence type bound on the excess risk and an interesting observation about strong convexity of the risk in the parameters of the network. The strong convexity here combined with Lipschitzness of the predictor allows to "convert" excess risk bound into the deviation between predictors.
The uncertainty estimation developed here is later used to give an explore-then-commit type of bandit algorithm with $T^{\\frac{2}{3}}$ regret.

**Summary Of The Review:**

The paper investigates a very relevant and interesting topic of uncertainty estimation in overparameterized neural networks. The interesting and novel part of the paper is in attempt to give uncertainty estimation for predictors rather than parameters. However, uncertainty estimation depends on the norm of a fitted network which is generally unclear in the interpolation setting. It is likely that this issue can be mitigated by analyzing the norm (e.g. given by GD solution -- right now no algorithm is considered for fitting the network), and for instance using some arguments as in Arora et al. 2019, however I feel that right now the entire picture is not very clear and the paper could benefit a lot from clarifying this.

---

> ### Author Response · Authors · 2021-11-23
> **Response to Reviewer LNcd**
>
> Thank you for your helpful comments and feedback!
>
> **Boundedness of parameters.**
> Our generalization bound does not depend on $\Vert\theta\Vert_F$. Instead, it only depends on $\phi_{\max}$, which is a bound on $\phi = \theta \theta^\top \in \mathbb{R}^{d \times d}$, i.e., $\Vert\phi\Vert_F \le \phi_{\max}$. Thus, our bound maintains a $\mathcal{O}(\frac{1}{\sqrt{n}})$ dependence even if $k$ increases with $n$, since $\phi_{\max}$ remains constant.
>
> **ETC bound.** As above, since our choice of $m$ only depends on $\phi_{\max}$, the value of $C_0$ and $C_1$ defined in Appendix C do not depend on $k$ or $T$. Thus, they do not grow with the sample size $n$.
>
> **UCB bound.** We can apply our approach to the UCB algorithms similarly, as we mentioned in our paper. We presented an ETC-type bound since it is a more direct application of our generalization bound; leveraging UCB is orthogonal to our contribution.

---

### Official Review · Reviewer_hpBY · 2021-11-02

**Correctness:** 4
**Technical Novelty And Significance:** 4
**Empirical Novelty And Significance:** Not applicable
**Recommendation:** 6
**Confidence:** 4

**Main Review:**

1. Despite that the authors have already mentioned that the analysis is focused on quadratic neural networks, the analysis of function identification seems to be very limited to quadratic neural networks and hard to generalize to other activations. This somehow constrained the potential extension of the proposed framework.
2. Identifying the functions are very different from identifying the parameters. Could the authors add a discussion to compare the pros and cons between them? Why function identification is a better definition besides the usage in over-parameterized networks? The authors mentioned that the analysis is valid for over-parameterized networks multiple times, but it is applicable simply because the formulation is very strong and avoids the parameters.
3. The authors should point out which bounds are newly derived and if there are existing bounds with similar settings, a discussion should be added to compare them.
4. The conclusion is too concise that it lacks a discussion on potential future works.

**Summary Of The Paper:**

The author proposed the concept of function identification to address the limitation of parameter identification in the over-parameterized setting. The function identification states that the outputs of quadratic neural networks with the empirically minimized parameters and the true parameters have bounded difference with high probability. The reason to adopt the definition of function identification analysis is that it is hard ti make parameter identification with neural networks which usually are over-parameterized. The main theoretical result is Theorem 3.6, which is directly used in Corollary 3.7 to derive a robust generalization error bound, i.e., the loss function is evaluated on an arbitrary distribution. The robust generalization error bound is then used for three cases: (1) the upper bound of the regret of quadratic neural bandits, (2) bounds for transfer learning, and (3) generalization bounds for neural module networks.

**Summary Of The Review:**

The consideration of function identification is an appealing tool for analysis since it only requests the outputs to be similar with high probability. The derivation then follows the analysis of quadratic neural networks. I think there are some discussion and comparisons needed (see bullet points above), and therefore I decided to hold my scores.

---

> ### Author Response · Authors · 2021-11-23
> **Response to Reviewer hpBY**
>
> Thank you for your helpful comments and feedback!
>
> **Generalization of activations.** We agree that studying other activation functions or deeper architectures is an important next step, but believe these are beyond the scope of our work. Importantly, on-distribution generalization of neural networks was first shown for simple models such as quadratic neural networks [1], before being extended to more complex architectures; we believe robust generalization will follow a similar trend. We have added a discussion of future work along these lines.
>
> **Function identification.**
> Our goal is to study settings where parameter identification cannot hold, such as overparameterized neural networks. We emphasize that we *prove* that function identification holds for quadratic neural networks, and that as a consequence, these models can generalize robustly. In other words, function identification is not an assumption we are making, but a desirable property that we are proving holds true.
>
> **Existing bounds.**
> To the best of our knowledge, all of our theorems are novel, and there has not been any work demonstrating robust generalization of neural networks (though they often leverage existing techniques). We have added a detailed discussion of technical novelty to Appendix A.
>
> **Conclusion.** Key directions for future work include extending our approach to new architectures. We have added a discussion of future works in this direction to our conclusion.
>
> [1] Du, Simon, and Jason Lee. "On the power of over-parametrization in neural networks with quadratic activation." International Conference on Machine Learning. PMLR, 2018.

---

### Decision · Program_Chairs · 2022-01-20

**Decision:**

Reject

**Comment:**

This paper tackles an interesting problem: distribution shift generalization often requires parameter identification but this is not possible for over-parameterized neural networks. This paper shows for quadratic neural networks, it is possible to identify the function without identifying the parameter.

This is an interesting result. However, reviewers raise concerns about the assumption and technical details. The meta-reviewer agrees with these concerns.